# Coconut genome assembly enables evolutionary analysis of palms and highlights signaling pathways involved in salt tolerance

Yaodong Yang [1,8], Stéphanie Bocs [2,3,4,8], Haikuo Fan[1,8], Alix Armero[3,8], Luc Baudouin [2,3,8 ✉],
Pengwei Xu[5,8], Junyang Xu[5], Dominique This [3], Chantal Hamelin[2,3,4], Amjad Iqbal [1], Rashad Qadri[1],
Lixia Zhou[1], Jing Li[1], Yi Wu[1], Zilong Ma[6], Auguste Emmanuel Issali[7], Ronan Rivallan[2,3], Na Liu[5], Wei Xia [1✉],
Ming Peng [6✉] & Yong Xiao [1✉]

Coconut (*Cocos nucifera*) is the emblematic palm of tropical coastal areas all around the globe. It provides vital resources to millions of farmers. In an effort to better understand its evolutionary history and to develop genomic tools for its improvement, a sequence draft was recently released. Here, we present a dense linkage map (8402 SNPs) aiming to assemble the large genome of coconut (2.42 Gbp, 2n = 32) into 16 pseudomolecules. As a result, 47% of the sequences (representing 77% of the genes) were assigned to 16 linkage groups and ordered. We observed segregation distortion in chromosome Cn15, which is a signature of strong selection among pollen grains, favouring the maternal allele. Comparing our results with the genome of the oil palm *Elaeis guineensis* allowed us to identify major events in the evolutionary history of palms. We find that coconut underwent a massive transposable element invasion in the last million years, which could be related to the fluctuations of sea level during the glaciations at Pleistocene that would have triggered a population bottleneck. Finally, to better understand the facultative halophyte trait of coconut, we conducted an RNA-seq experiment on leaves to identify key players of signaling pathways involved in salt stress response. Altogether, our findings represent a valuable resource for the coconut breeding community.

[1] Hainan Key Laboratory of Tropical Oil Crops Biology/Coconut Research Institute, Chinese Academy of Tropical Agricultural Sciences, 571339 Wenchang, Hainan, P. R. China. [2] CIRAD, UMR AGAP, F-34398 Montpellier, France. [3] AGAP, Univ. Montpellier, CIRAD, INRAE, Institut Agro, F-34398 Montpellier, France. [4] South Green Bioinformatics Platform, Bioversity, CIRAD, INRAE, IRD, F-34398 Montpellier, France. [5] BGI Genomics, BGI-Shenzhen, Shenzhen 518083, P. R. China. [6] Institute of Tropical Bioscience and Biotechnology, Chinese Academy of Tropical Agricultural Science, 571101 Haikou, Hainan, P. R. China. [7] Station Cocotier Marc Delorme, Centre National De Recherche Agronomique (CNRA)07 B.P. 13, Port Bouet, Côte d'Ivoire. [8] These authors contributed equally: Yaodong Yang, Stéphanie Bocs, Haikuo Fan, Alix Armero, Luc Baudouin, Pengwei Xu. ✉email: luc.baudouin@cirad.fr; saizjxiawei@hainu. edu.cn; pengming@itbb.org.cn; xiaoyong1980@catas.cn

Coconut (*Cocos nucifera* L.) is a continuously fruiting evergreen perennial tropical monocotyledon, adapted to humid and sub-humid coastal environments. Because of its importance in sustaining the life of coconut growers and its various economic uses, coconut is often regarded as a "tree of life". It is grown in more than 92 countries, covers 12.2 million ha of plantation area, of which 85% is found in the Asia-Pacific region. Global coconut oil production was 3.1 million tons in 2014 (the last available figure in FAOSTAT). Coconut belongs to the palm family (Arecaceae) and is the only species of the genus *Cocos*. The oldest known *Cocos* fossils date from the Paleocene period (around 56–66 million years ago) and were found in India[1,2], and in the Eocene in Australia. The coconut lineage appears thus to be notably older than the 23.9–44.4 Mya range proposed by Meerow[3] for the divergence from *Attalea* while Xiao et al.[4] find a broad 25.4–83.3 Mya range for the divergence with *Elaeis*. Most coconuts are cross-pollinating and fast-growing (talls). Self-pollinating coconuts (dwarfs), which grow more slowly, appeared in South-East Asia and appear to be the result of a domestication process[5].

Coconut is naturally adapted to the coastal environment and, as a facultative halophyte, is tolerant to variable levels of water salinity. In Cl-deficient soils, salt can be used as fertilizer, and a 3.8 kg/tree/year NaCl application results in more than doubled copra yield[6]. Fertilization using seawater is also being considered in the Hainan Island province of Southern China for the local variety (Hainan tall). Dwarf genotypes, believed to be derived from tall types through domestication[5], are considered to be more susceptible to environmental stresses like salt. Therefore, understanding the relevant mechanisms will contribute to the breeding of salt-tolerant coconut varieties, complementing the research on the adaptation mechanism of salt stress in other crops.

In our previous publication of the draft coconut genome[4], we investigated the evolution of transporter gene families in coconut. In the present work, we were more interested in identifying gene regulatory networks of the signaling cascades controlling response to salt stress. Salinity stress causes both osmotic stress and ionic stress that inhibits normal plant growth and cell division[7]. At the physiological level, Munns and Tester[8] observed that plants respond to salinity in two phases: a rapid, osmotic phase that closes stomata, inhibits the growth of young leaves, and a slower, ionic phase that accelerates senescence of mature leaves. During the fast response (early molecular signaling and quiescent growth phase, probably within seconds to hours in root[9] and minutes to hours in leaves[8]), the plant reacts with the protein arsenal in place (e.g. post-translational modification during early signaling); whereas, during the acclimation slow response (growth recovery phase, probably within days to weeks), gene expression modulations have induced changes at the proteome content level[10]. In any case, both osmotic and ionic stresses appear to be sensed during the rapid signaling of the salt stress[11]. The loss of turgidity caused by osmotic stress is perceived by the turgor-sensing mechanosensitive receptor kinase-cyclase[9,12] and the early transducers can be cGMP, $Ca^{2+}$, and reactive oxygen species (ROS)[9,12,13]. Both ROS and $Ca^{2+}$ modulate the biosynthesis of abscisic acid (ABA)[9] and ABA signaling plays an important role in the abiotic stress response and tolerance of plants[14] (e.g., osmotic stress). While the mechanisms of osmotic stress are well known in plants, little is known about the mechanisms of ionic stress: for instance, what is the sodium ion entry point[15] and how is $Na^+$ sensed in the cell[16]?

A genome draft of coconut (cultivar Hainan Tall or HAIT) was released two years ago[4] and generated around 2.20 Gbp of sequence scaffolds, i.e. ~91% of the coconut genome and 28,039 annotated protein-coding genes. The coconut genome is large (estimated as 2.42 Gbp through to Kmer analysis[4]), 34% larger

than the oil palm genome (1.8 Gbp) and 3.6 times larger than date palm (0.67 Gbp). In plant genomes, large size and structural variation even among closely related species is often due to differences in their history of polyploidization[17] and/or amplification of long terminal repeat retrotransposons[18–21] (LTR-RTn). It has been shown that 67.1% of the coconut genome consists of LTR sequences. LTR retrotransposons are major components of plant genome modification and reorganization[22–24]. However, the nature and dynamics of changes of LTR-RTn contents during genome expansion and speciation are still poorly understood[25].

Correlatively to its large size, the coconut genome is richer in repeated sequences than is oil palm (73% and 57%, respectively), resulting in a more fragmented draft genome (N50 of scaffold lengths 418 kb and 1045 kb, respectively). Accordingly, ordering the scaffolds represents a strenuous task, which demands a high-density linkage map.

Another draft sequence (cultivar Catigan Green Dwarf or CATD) was published recently[26]. The quality of this sequence is attested by a somewhat larger N50 and some improvement in genome assembly and annotation completeness (BUSCO[27] score). It differs from the results on the HAIT by a smaller estimated genome size (2.15 Gbp vs 2.42), a smaller proportion of predicted LTR-RTns (60.3%), and a higher proportion of protein-coding genes (34,958 vs 28,016). This difference appears to reflect differences in assembling strategies rather than biological differences between CATD and HAIT. In fact, flow cytometry reveals intraspecific variation but to a much lesser extent and tends to yield larger values than does Kmer analysis (around $C = 2.9$ Gbp[28] or $C = 2.7$ Gbp[29]). The CATD assembly also identified 25% more genes, resulting partially from a larger proportion of duplicated genes but also from a larger number of unique genes, especially among the smaller genes (Supplementary Fig. S1).

In the present work, we generated a linkage map using the genotyping by sequencing (GBS) technology based on a backcross mapping population and used it to anchor sequence scaffolds onto 16 coconut pseudochromosomes. This allowed an in-depth comparison with related genomes, providing insights into palm genome evolution and genome expansion in coconut due to transposable elements (TEs). We also analyzed the expression level of transcripts involved in signaling pathways through a salt-stress RNA-seq experiment on leaves of Hainan tall and aromatic dwarf seedlings. This allowed us to perform the first multi-omics comparison providing insights into palm transcriptome regulation through the identification of key genes differentially expressed during the early signaling of salt stress in coconut leaves. Along with previously released transcriptomes[30,31], this linkage map-assembled genome provides an invaluable resource for coconut genetic improvement and understanding of the mechanisms underlying key traits such as oil production, economically interesting variants of the albumen texture[32], disease resistance, and abiotic stress tolerance.

## Results and discussion

**Genetic mapping of a coconut bi-parental population by GBS.** In order to assign the coconut sequences to their positions on coconut chromosomes, we created a linkage map, based on a back-cross MYD × (MYD × WAT) produced and planted in Côte d'Ivoire (WAT stands for the West African Tall and MYD for the Malayan Yellow Dwarf). After DNA extraction, a legitimacy test involving fifteen microsatellite markers was performed on 320 progenies and confirmed the origin of 292 progenies, which were genotyped by GBS, along with the parents. We obtained GBS sequences from 240 progenies. The 9,459,318 variants assigned to 7904 scaffolds were initially identified by TASSEL-GBS[33] pipeline and VcfHunter package[34]. Several filtration steps (see "Methods")

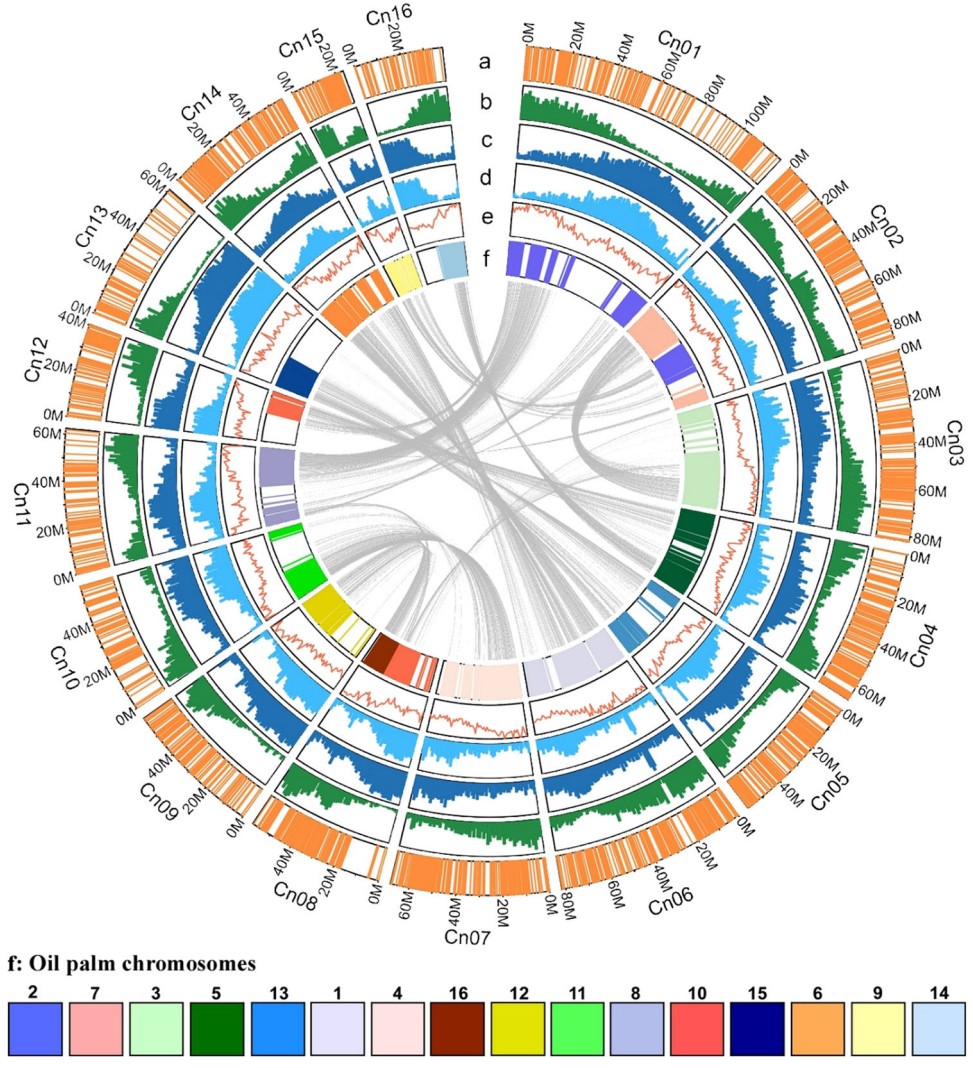

**f: Oil palm chromosomes**

**Fig. 1 Circos of the genome features and ancestral blocks post-p WGD.** Concentric circles show aspects of the genome. a Distribution of GBS markers. b Gene number per window. Min: 1, max: 59. c Repetitive sequence (SSR excluded) percentage per window. Min: 28%, max: 98%. d LTR percentage per window. Min: 28%, max: 98%. e GC content per window. Min:33%, max:40%. f Orthologous coconut polypeptides are painted according to the 16 oil palm chromosomes (the chromosome number is indicated above the color boxes). The inner links represent coconut paralogy resulting from the p WGD. For tracks b–d, the window size is 1 Mb.

retained 8402 SNPs with less than 20% missing data per marker, assigned to 2303 scaffolds, and scored on 216 progenies with <25% missing data.

Based on recombination distances, Joinmap[35] identified 16 linkage groups, corresponding to the number of chromosome pairs in coconut. Only one of them (Cn15) was made up of two weakly linked subgroups (see below). Software Scaffhunter[36] was used to order markers within linkage groups while preserving scaffold integrity. We could also verify that appreciable linkage occurred only within the linkage groups identified by Joinmap (Supplementary Fig. S2). After genotyping error correction, the estimated chromosome length after error correction varied from 81 cM (Cn13) to 225 cM (Cn06), yielding a total length of 2365 cM. The number of markers per chromosome varied from 244 (Cn15) to 904 (Cn01), achieving a dense coverage of most of the coconut genome (Fig. 1a). In average, the marker density was 8.6 SNPs per Mbp.

**Segregation distortion analysis.** Taking into account all chromosomes but Cn15, the percentage of heterozygotes was 47.77%

± 4.16%, close to the expected value (50.00%). Contrastingly, chromosome Cn15 presented a strong heterozygote deficit with an average percentage of only 27.5%.

Heterozygosity was maximal (≈40%) toward the ends of the chromosome and decreased linearly toward the middle (14.20%), where no markers could be identified, due to insufficient polymorphism, generating a 10.36 cM gap (Fig. 2). This is the signature of selection in favor of the pollen grains holding the allele inherited from the MYD parent at a single locus located at position 44.5 cM. The selection at the post-zygotic stage is ruled out because coconut dwarf × tall hybrids are not difficult to produce[37], are high producing and nuts germinate normally[38]. Moreover, such pollen–pistil interaction was already observed by Sangaré in a pollen mixture experiment based on Mendelian shoot color traits[39] showing that pollens from red and yellow dwarfs applied on yellow dwarf female flowers are equally efficient, but appreciably more efficient than pollen from a green or brown tall. This result differs only from ours by the selection intensity: the fitness of the pollen with the "tall" allele (0.58) is higher than in our case (0.113). The most parsimonious explanation for this difference would involve interaction between

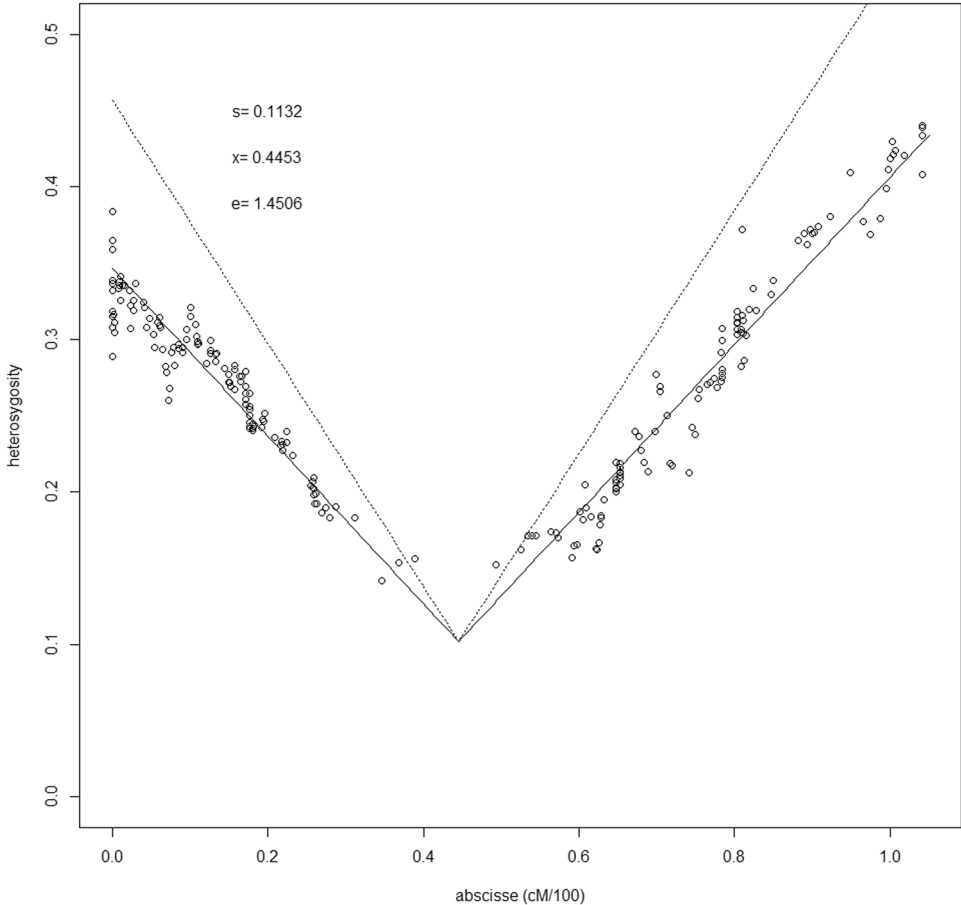

**Fig. 2 Graphical estimation of the position of a locus under selection on LG Cn15.** Each dot represents one locus. Expected variations of heterozygosity along the chromosome are modeled by the oblique lines (dotted lines, without genotyping errors correction, solid line, with error correction). s = fitness of the heterozygote genotype; x = position of the selected locus; e = correction factor.

the above nuclear locus and a hypothetical cytoplasmic gene. The lack of markers around this gene prevented us from assembling this portion of the coconut sequence, however, based on the homology between Cn15 and chromosome 9 of *Elaeis guineensis* (see below), we determined that 18 coconut scaffolds are likely to cover 9 Mbp in the missing portion (Supplementary Data 3). The gene involved in competition is probably among the 209 protein genes they contain. Incidentally, the estimation procedure determined that the estimated length of the chromosome is 1.45 times the actual length. If this ratio can be extended to the whole coconut genome, the actual linkage map length would be only 1631 cM.

This pollen competition phenomenon may be interpreted in two ways. Either it is part of the domestication syndrome in dwarf coconuts[5] or it is the result of isolation between coconuts from the Indian and Pacific Oceans due to glaciations at the Pleistocene and the subsequent lowering of sea level[40]. In the first case, pollen competition would represent a novel factor contributing to uniformity in dwarf coconuts cultivars, in addition to the coincidence of male and female flowering phase[41,42]. In the second case, it would result from a tendency toward allopatric speciation. Deciding between these hypotheses would require more mixed pollen experiments where pollen from Pacific talls would be applied to dwarfs.

**Scaffold anchorage onto the map and quality assessment.** The 2303 anchored scaffolds represented only 2.03% of the scaffolds but 46.6% of the total genome and 77.1% of the proteins (Table 1). In spite of the large genome size of a coconut and of its richness in repeated elements, the results we obtained are comparable to those obtained in oil palm[43] in terms of the proportion of the genome covered by the 16 pseudochromosomes (46.6% and 43%, respectively) as well as of the size of unique sequences (between 0.7 Gbp and 0.8 Gbp). However, this required a much larger number of scaffolds (2303 instead of 304).

As in most plants, coconut chromosome architecture is shaped by gene-rich and repeat-rich regions also known as euchromatin and heterochromatin, respectively[44]. Heterochromatin is located toward centromeres, telomeres, and chromosome fusion footprints (Fig. 1 and Supplementary Fig. S3). The abundance of transposable elements accounts for the fact that <50% of the coconut genome could be assembled into pseudomolecules. Recombination is suppressed in heterochromatin and fewer markers could be obtained. While the exact boundaries of euchromatin and heterochromatin are difficult to define, regions with a recombination rate usually ranging from 2 to 5 cM/Mb, with typically 10–20 markers per Mb and 20–50 protein genes per Mb can safely be assigned to euchromatin. This is large enough to ensure fairly accurate reconstruction of such regions, omitting only numerous, but very small scaffolds. Contrastingly, in heterochromatin, there are less than five markers per Mb in the pseudomolecule, resulting in incomplete assembly. Moreover, the recombination rate drops to 1 cM/Mb or less, leading to an uncertain ordering of the scaffolds. Nevertheless, most of the euchromatin was assembled; it represents at least 57% of the pseudomolecules and 80% of their proteins).

**Table 1 Summary statistics of the assembled coconut sequence.**

| Chromosome | Total length | Scaffold count | Ungapped length | N50 | L50 | Spanned gaps | Unspanned gaps | Gene number | GC% (ungapped) | Length (cM) |
|---|---|---|---|---|---|---|---|---|---|---|
| Cn01 | 119,963,539 | 188 | 117,957,365 | 1,503,304 | 25 | 1,987,374 | 18,800 | 2277 | 37.5 | 203 |
| Cn02 | 85,636,561 | 204 | 83,804,978 | 753,179 | 28 | 1,811,283 | 20,300 | 1997 | 37.6 | 183 |
| Cn03 | 82,374,452 | 202 | 80,663,866 | 744,939 | 29 | 1,690,486 | 20,100 | 1705 | 37.7 | 206 |
| Cn04 | 66,811,639 | 175 | 65,401,075 | 779,691 | 23 | 1,393,164 | 17,400 | 1610 | 37.7 | 174 |
| Cn05 | 57,731,914 | 137 | 56,647,193 | 642,592 | 25 | 1,071,121 | 13,600 | 1104 | 37.2 | 138 |
| Cn06 | 81,752,308 | 253 | 79,872,242 | 691,323 | 34 | 1,854,866 | 25,200 | 2095 | 38.2 | 225 |
| Cn07 | 68,107,953 | 219 | 66,405,264 | 548,297 | 34 | 1,680,889 | 21,800 | 1583 | 37.8 | 179 |
| Cn08 | 58,726,133 | 155 | 57,524,819 | 657,407 | 14 | 1,185,814 | 15,500 | 1214 | 37.6 | 111 |
| Cn09 | 59,467,968 | 107 | 58,339,163 | 933,531 | 21 | 1,118,205 | 10,600 | 1026 | 37.3 | 98 |
| Cn10 | 57,291,198 | 101 | 56,367,391 | 1,522,480 | 11 | 913,807 | 10,000 | 1066 | 37.3 | 131 |
| Cn11 | 62,458,273 | 107 | 61,301,060 | 1,391,469 | 14 | 1,146,613 | 10,600 | 1379 | 37.5 | 177 |
| Cn12 | 40,247,383 | 90 | 39,443,120 | 813,247 | 15 | 795,363 | 8900 | 868 | 37.7 | 87 |
| Cn13 | 61,684,062 | 70 | 60,924,034 | 1,851,699 | 9 | 753,128 | 6900 | 847 | 37.1 | 81 |
| Cn14 | 56,264,922 | 157 | 55,130,168 | 715,325 | 24 | 1,119,054 | 15,700 | 1309 | 37.7 | 181 |
| Cn15 | 23,504,088 | 75 | 22,734,417 | 508,623 | 13 | 762,271 | 7400 | 646 | 37.8 | 104 |
| Cn16 | 38,224,781 | 63 | 37,575,163 | 1,129,175 | 10 | 643,418 | 6200 | 867 | 37.8 | 86 |
| All anchored sequences | 1,020,247,174 | 2303 | 1,000,091,224 | 884,163 | 293 | 19,927,250 | 228,700 | 21,593 | 37.6 | 2365 |
| Not anchored | 1,098,384,151 | 26,453 | 1,147,115,255 | 682,484 | 178 | 35,092,692 | | 5999 | 37.0 | |
| All sequences | 2,202,455,121 | 113,653 | 2,147,206,479 | 418,659 | 1055 | 55,019,942 | 228,700 | 28,039 | 37.0 | |
| Percentage anchored | | 2.03% | 46.6% | | | | | 77.1% | | |

An improved version of the coconut sequence requires combining mapping data with other methods such as Hi-C and optical mapping. Such methods are required to accurately assemble heterochromatin. The CATD sequence will help to decide between hypotheses. In any case, the present linkage mapping will play an essential role. In fact, while Hi-C is generally more efficient at a short distance, it is based on the 3D arrangement of the nucleus, occasionally resulting in chimeric assembly as shown in Supplementary Fig. S4.

In most of the chromosomes (e.g. Cn01), genes are abundant toward the extremities and less frequent in between, indicating the presence of the centromere (Fig. 1b). Gene density ranged from 1 to 59 gene/Mb, similar to what is observed e.g. in peach[45] and jujube[46]. The pattern presented by transposable elements (TE) is opposite to that of genes. They mostly consist of LTR retrotransposons (Fig. 1c, d). TEs tend to accumulate toward centromeres and a few chromosomes appear to be acrocentric (Cn09, Cn12, Cn13, and Cn16). Figure 1f represents synteny with oil palm. Orthologous genes were found in all chromosomes. Sectors left blank corresponds to TE-rich regions where orthologs are rare or absent. Cn02 and Cn08 present evidence of chromosome fusions, parts of them being homologous to different oil palm chromosomes, and peaks of TE are located precisely at the points where chromosome fusion took place. Cn15, already mentioned, presents an irregular pattern. The GC content is distributed unevenly in most pseudochromosomes and tends to be higher in gene-rich regions (Fig. 1e). Paralogy within the coconut genome is represented by the inner gray lines.

**Whole-genome duplications and the origin of coconut palm.** An analysis of the size and frequency of paralogous gene families accumulated in the coconut genome provided a record of the whole-genome duplication process. Based on results obtained in other palm species[47,48], we expected to find extensive homology resulting from two whole-genome duplication (WGD $p$ and $\tau$). We were able to verify this using Ks and 4DTv analyses. By aligning 28,039 coconut gene sequences among themselves and performing Ks analysis, we were able to represent paralogy in a dot plot showing extensive segmental duplication between coconut pseudochromosomes (Supplementary Fig. S5). The distribution of Ks presented two main peaks and by comparison with other model plants, we could assign the first one around Ks ≈ 0.32 to WGD $p$ shared by palms and the second one, around Ks ≈ 0.88, to WGD $\tau$ shared by commelinids, which are represented in Supplementary Fig. S5a by pink dots and blue dots, respectively. The erosion of the synteny is more pronounced in the latter case, which confirms that it is the oldest one. This allowed us to identify 2666 paralog protein pairs, forming 85 paralogous gene blocks (Supplementary Data 1) that could be grouped further into 13 blocks according to the chromosome pairs involved.

We further studied the distribution of 4DTv (a subset of synonymous mutations) between the coconut and oil palm genomes as well as within genomes. Based on 2490 paralogous gene pairs in 260 collinear regions of the coconut genome, we assessed the distribution of 4DTv distances among coconut genes, which showed two distinct peaks: 4DTvB ≈ 0.125 and 4DTvB ≈ 0.358 (Fig. 3a). The distribution in oil palm was very similar. In the same way as with Ks, these can be attributed respectively to WGD $p$ and to WGD $\tau$. With both Ks and 4DTvB, the ratios of the distances for WGD $p$ and WGD $\tau$ is ~2.8. Distances between coconut and oil palm homologs exhibited an additional peak at 4DTvB ≈ 0.05. This peak would correspond to a speciation event involving the most recent common ancestor of coconut and oil palm, resulting in the ancestral lines of Attaleineae and Elaeidineae, respectively.

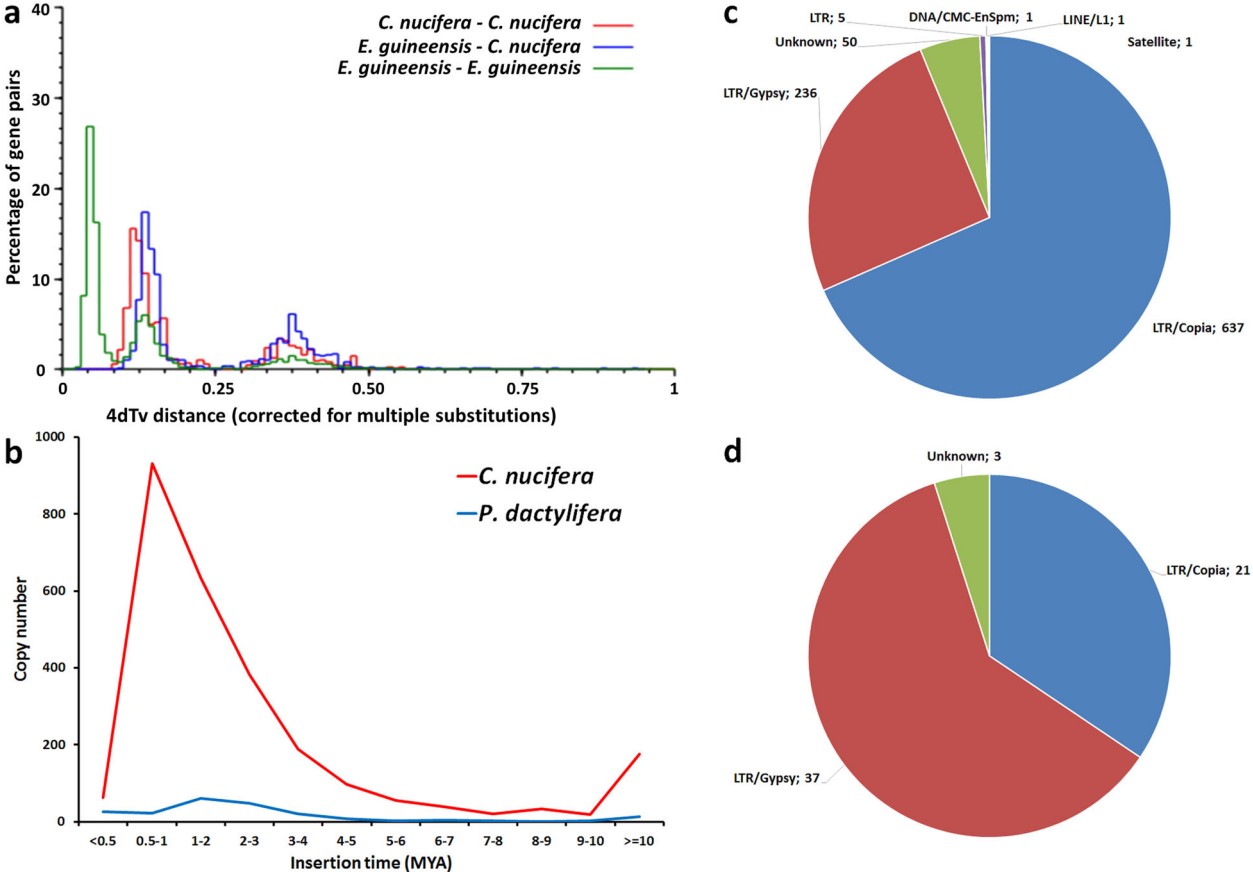

**Fig. 3 Comparison between the genomes of coconut and of related species. a** Distribution of 4DTV distances between homologs in coconut, in oil palm, and between these species. **b** Distribution of copy numbers of LTR over the insertion time in coconut and date palm, showing a peak in coconut in the last MYA. **c** LTRs composition at peak insertion time for coconut. and LTRs composition at peak insertion time. **d** Same as (**c**) for date palm.

Once we had evidence for WGD $p$ in coconut, we were able to identify synteny blocks at the chromosome level, which allowed us to identify the ancestral blocks pre-dating this event. A putative ancestral pre-$p$ genome of palms was previously proposed, based on 13 large blocks of duplication identified in oil palm chromosomes[49]. We used the orthology relationships between coconut and oil palm to identify the same blocks in the coconut genome.

In total, 9866 orthology relationships between coconut and oil palm allowed identifying these ancestral blocks in the coconut genome (Supplementary Data 2). Out of the 2666 coconut paralogous pairs, oil palm orthologous genes were retrieved for both members in 889 cases, confirming block orthology in all cases but two. Combining automatic analysis with the MGRA[50] software and manual curation, we propose an ancestral genome pre-$p$ of 9 chromosomes made of 13 duplicated blocks (from A to M) in coconut as follows: (1) M, (2) A, (3) G, (4) L, (5) C, (6) EF, (7) JK, (8) BD and (9) HI (Supplementary Fig. S6). This corresponds to "scenario 2" proposed in Figure 10 of Armero's PhD thesis[51]. We identified structural modifications between the pre-$p$ ancestor (Fig. 4a) and the modern genomes of coconut and oil palm (Fig. 4a–c). The post-$p$ ancestor ($n = 18$) of *Cocos* and *Elaeis* experienced at least four fissions (E–F, K–J, B–D, H–I) and four fusions (E–J, K–F, B–JK, D–H) before they diverged (Fig. 4a). In the process of speciation, the coconut and oil palm ancestral lines independently underwent two additional fusions (Fig. 4b), accounting for the modern structure of coconut and oil palm genomes ($n = 16$ in both species) but different chromosome pairs were involved: Cn02 (EJ-C) and Cn08 (FK-L) in coconut, Chr02

(BD-C) and Chr10 (FK-HD) in oil palm (Fig. 4c). This scenario is detailed in Fig. 5a and the resulting synteny with oil palm in Fig. 5b and Supplementary Fig. S7. Our results contribute to enrich our understanding of the evolutionary history of palms, as presented in Fig. 1 of Murat et al.[49]. A detailed account of the evolutionary history of palm genomes would require studying more palm species (ideally, one per subfamily) and an outgroup (e.g. banana, a member of Zingiberales, which are a sister taxon of Arecales).

Our results are compatible with those of the CATD study[26] if we equate their β event with the τ WGD and their α event with the $p$ WGD. In fact, *Musa acuminata* has the former but not the latter. They also identify an earlier γ event, which could be the ε WGD, common to angiosperms[49]. The $p$ and τ WGDs were confirmed by a new study covering a wide range of palm transcriptomes[52]. However, that study suggests a $2n = 30$ chromosome common ancestor of palms, which is not supported by chromosome-scale synteny. In fact, two WGDs account for the chromosome numbers: starting from the common ancestor of flowering plants (AMK)[49] with $n = 5$, we obtain $n = 10$ after WGD τ. As shown in Fig. 5, a chromosome fusion and WGD $p$ led to a $n = 18$ ancestor and finally to $n = 16$ in both coconut and oil palm.

**TE burst and genome expansion.** Plant genomes are exceptional for their great variation in genome size, which results from two main factors: the polyploidy level and the abundance of non-coding DNA, especially the contribution of transposable elements

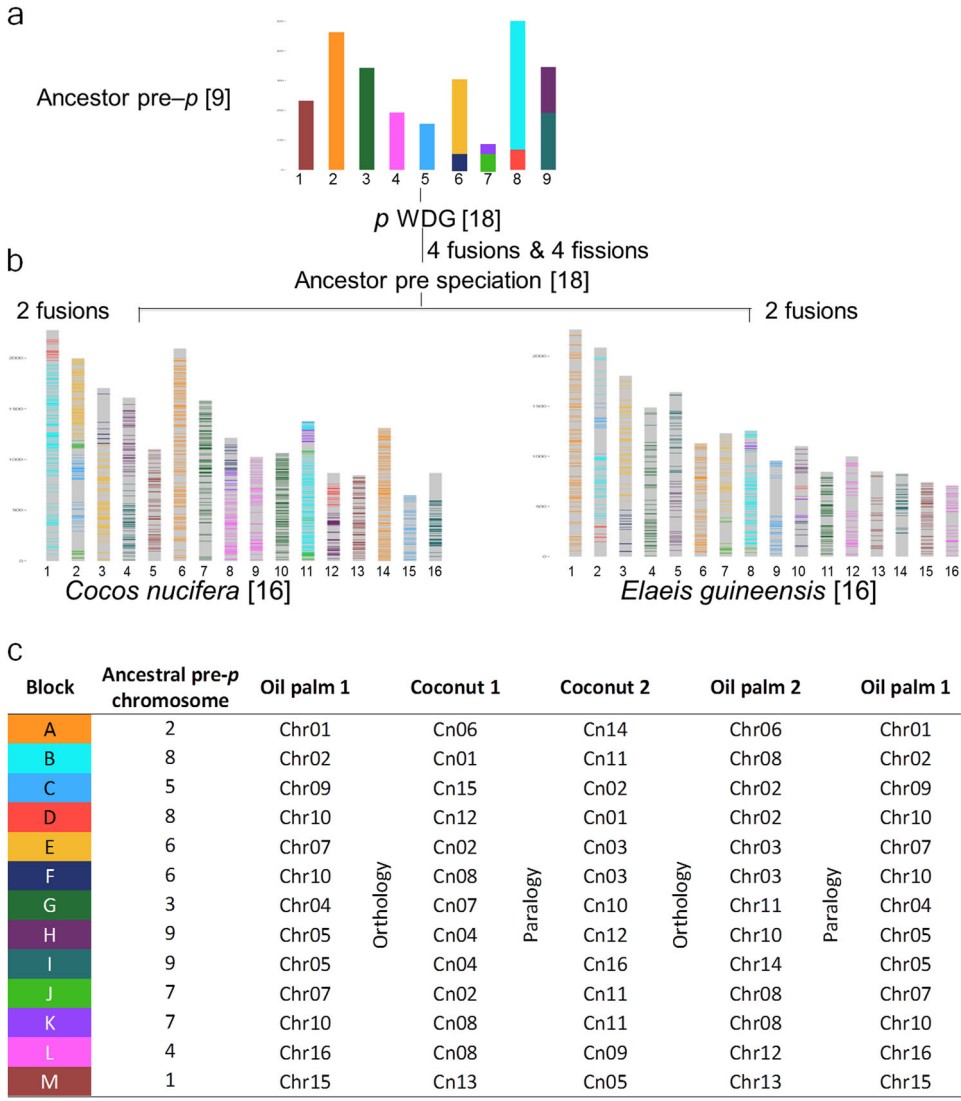

**Fig. 4 Schematic representation of homology in coconut and in oil palm. a** schematic representation of the nine chromosomes of the hypothetical pre-*p* ancestor in terms of 13 ancestral blocks A to M based on segmental homology within and between the oil palm and coconut (color code in the table below). **b** Location of 9866 homologous genes on the *Cocos* and *Elaeis* chromosomes. Duplication of the ancestral chromosome 2 resulted in chromosomes 6 and 14 in coconut, homologous to chromosome 1 and 6 of oil palm (block A). Other chromosomes derive from two ancestral chromosomes due to rearrangements (see details in Fig. 5). **c** The 13 ancestral blocks and their locations in the pre-*p* chromosomes and in both palm species. Each genome is split into two sub-genomes resulting from WGD *p*. Sub-genome 1 of oil palm is repeated to the right to show paralogy relationships within oil palm. (modified after figure 9 of Armero 2017[51]).

(TEs), including retrotransposons. The coconut genome is the largest among the palm species that have been sequenced so far and the richest in LTR retrotransposons. The proportion of LTR-RTns is 57% in oil palm[43] and 22% in date palm[53]. It has been shown that 72.75% of the coconut genome is represented by TEs, of which 92.23% are LTR-RTns[4].

To investigate the genome expansion event in coconut, we dated the insertion time of all LTRs based on divergence analysis[18]. The LTR insertion times were calculated in *C. nucifera* and *Phoenix dactylifera* using the LTR_STRUC program[54] and their distributions are shown in Fig. 3b. In coconut, the insertion rate was low as of 6 million years ago, increased gradually, and reached a peak at 0.5–1 Mya, suggesting that the expansion of the coconut genome was quite recent, in comparison with the age of the Arecaceae family. In addition to the differences in LTR insertion rates, their natures differ: in coconut, 67% of LTR retrotransposons in the insertion peak are in the *Copia* subfamily, while the *Gypsy* subfamily dominates in date palm, representing 61% of the LTRs (Fig. 3c, d).

While the CATD sequence[26] appears to exhibit a smaller proportion of LTRs and a lower *Copia/Gypsy* ratio, the chronology of the LTR invasion is very similar, at least if we rely on the Ks values. Although the substitution rate in the CATD study is twice the one we adopted, both studies agree on the occurrence of a massive and recent LTR-RTn invasion followed by reduced activity. We can speculate on the causes.

LTR-RTns are major constituents of the plant genomes and are largely responsible for genome size variation, genome differentiation, and perhaps speciation[55]. Other examples range from *Arabidopsis thaliana* (~157 Mb) with 5.60% of LTR-RTns[56], to rice (~389 Mb) with ~22% LTR-RTn sequences[57], and maize (~2.3 Gbp) with 74.6% LTR-RTns[58]. In conifers with a genome

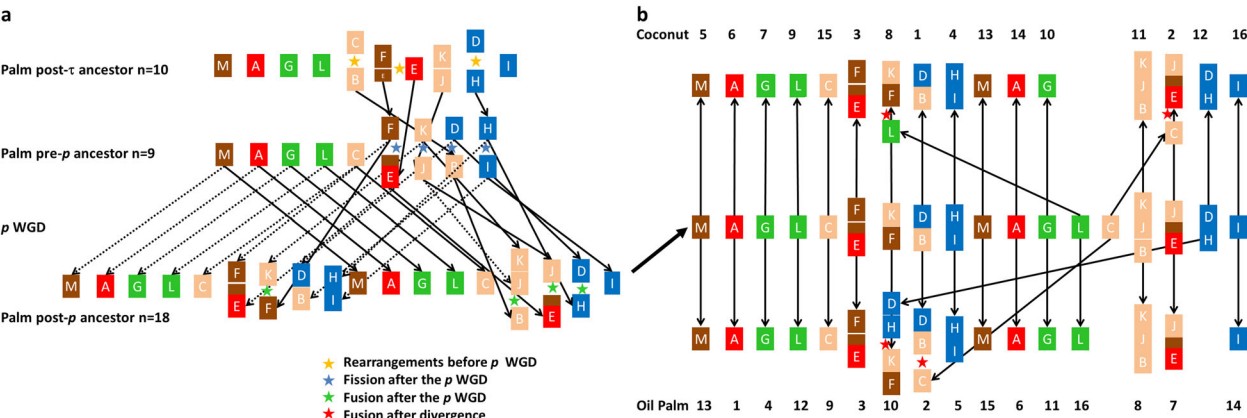

**Fig. 5 Hypothetical scenario of palm genome evolution. a** From WGD τ to the last common ancestor of coconut and oil palm ($n = 18$). The synteny blocks are colored according to their putative origin in the common ancestor of flowering plants ($n = 5$) in Murat et al.[49]. **b** Divergence of coconut and oil palm from their last common ancestor, resulting in their $n = 16$ current genomes by two fusions in each species.

**Table 2 Distribution of coconut genes according to expression pattern during salt stress and annotated to signaling pathways.**

| Putative salt response genes | | **56** | **4** | **22** | **20** | **50** | **152** |
|---|---|---|---|---|---|---|---|
| **DE or CHE genes** | | **Assigned to pathway** | | | | | |
| | Expression pattern | ABA | ABAi | Ca$^{2+}$/PLC | ROS | SOS | Total |
| 194 | CHE | 5 | – | – | 5 | 1 | 11 |
| 98 | Ea+ | 4 | – | – | 1 | – | 6 |
| 16 | Ea+La+ | 1 | – | – | – | – | 1 |
| 188 | La+ | 11 | 1 | – | 3 | 3 | 19 |
| 77 | Ea− | 6 | – | – | – | 1 | 7 |
| 49 | Ea−La− | 4 | – | – | 1 | 1 | 6 |
| 140 | La− | 7 | – | 1 | 5 | – | 13 |
| 17 | Ea−La+ | 3 | – | – | – | 1 | 4 |
| 7 | Ea+La− | – | – | – | – | – | – |
| 786 | Total | 41 | 1 | 1 | 15 | 7 | 65 |

The first column is devoted to the distribution of the eight genes classes with a significant pattern. In the second column, "Ea" refers to increased (+) or decreased (−) expression at 4 h, "La" to variations of expression at 6 or 10 days. "CHE" stands for "constitutively highly expressed. The first row is devoted to the distribution of the 152 homologs of rice or Arabidopsis genes involved in salt resistance. The rest of the table represents the distribution of the 65 genes that are common to both sets.

size range from 18 Gbp to over 35 Gbp for most species, the accumulation of LTR retrotransposons is assumed to account for their very large genomes, and this was confirmed by recent genome sequence analyses[59–61]. LTR retrotransposons have been shown to be responsible for wide genome expansions[18,62,63] and appear to have undergone recent amplifications within the past 15 million years[18,25,64]. In the case of coconut, the expansion peak was intense and much more recent, culminating around 0.5–1 Mya. The dynamics of TE invasions appear to depend on factors such as transposition rates, gene regulation, and host population dynamics[65]. An abundance of TEs in itself can be regarded as a disadvantage and tends to be eliminated by natural selection. But TEs contribute to horizontal gene transfers and can affect genome regulation, possibly contributing to greater adaptability. For example, it has been proposed that TE expansions could accompany population bottlenecks[66]. In this case, the natural selection pressure against the detrimental effects of TEs becomes insufficient to prevent an invasion. The authors also speculate on the possibility of positive effects: TEs modifying genome expression could promote better adaptability to stress or to a changing environment[67,68]. This seems to apply remarkably to coconut. From the beginning of the Pleistocene (2.6 My) until the end of this period, some 20 glaciation periods occurred in succession, resulting in fluctuations in sea level extending to more than 100 m. This means that at the height of these variations, the coastline could move forward or backward by tens of meters

during the life of a coconut tree. As coconut depends on the seaside to disseminate its seeds, the effects must have been dramatic. However, atolls[69] may have offered them a refuge: coral, as a living organism, naturally adjusts to sea level, offering coconut tree a much more stable environment, but of a smaller size than continental margins.

**Signaling pathways involved in salt-stress response**. As noted above, coconut uses the ocean currents to disseminate its seeds and its natural habitat consists of seashores and estuaries where freshwater and seawater mix. This means that it must be adapted to a variable, but often high water salinity. In order to understand the response of coconut to salt-induced stress, we submitted young plants from two varieties (Hainan tall and Aromatic dwarf) to a salt-stress experiment. We collected leaf RNA samples from tall and dwarf coconut genotypes at the following times: 0 h, 4 h, 6 days, and 10 days after saltwater application. RNA was extracted and sequenced. Incidentally, this allowed cross-validation of the quality of the draft sequence and of the transcriptome by aligning the clean RNA-seq reads onto both the scaffold reference and the annotated gene reference (see Supplementary Data 4 and 5). The overall mapping rate of 86% for the scaffolds (50% for gene reference), suggesting that the genome is largely complete[70]. In all, 23,651 genes (84% of annotated genes) showed a complete time-course RNA expression profile

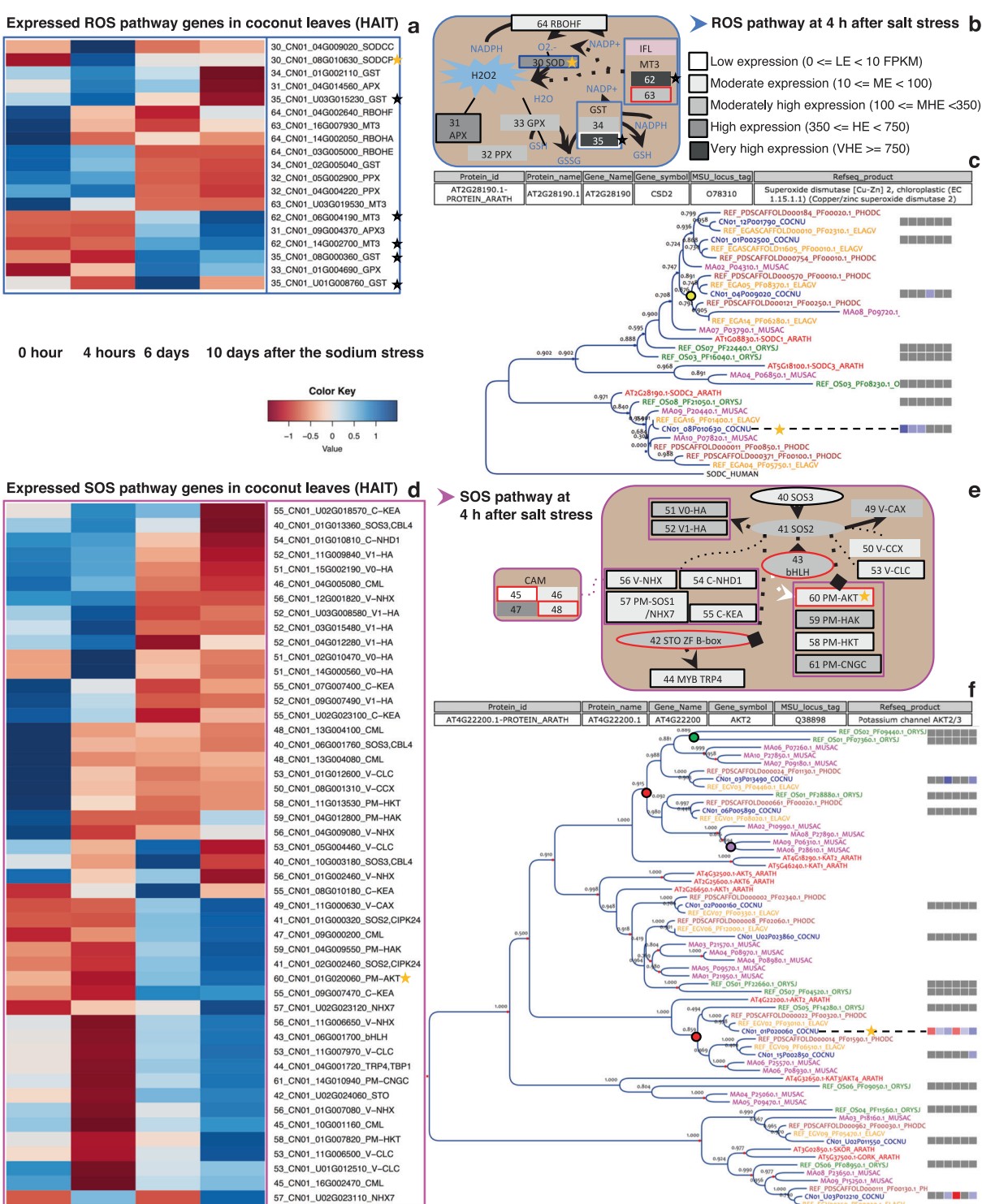

**Fig. 6 Early gene regulation during salt stress in coconut leaves. a** Normalized transcriptional expression level in Hainan tall leaves at time points 0, 4 h, 6 days, and 10 days after salt stress for the DE and CHE genes of the ROS pathway. Each gene is identified by its expression class number (ECN), gene identifier, and function symbol. **b** Schematic representation of the ROS signaling pathway at 4 h (see Supplementary Fig. S9 for more details). **c** Gene phylogeny of the SODC family across coconut (COCNU), date palm (PHODC), oil palm (ELAGA), banana (MUSAC), japonica rice (ORIZJ), and Arabidopsis (ARATH). Colored bullets represent WGDs (yellow, red, purple, and green for respectively p, τ, α/β/γ, and ρ/σ WGDs). Expression differential of coconut genes is displayed with blue and red squares for up and downregulation, respectively. The first three columns are for the Hainan tall variety 4 h, 6 days, and 10 days) and the last three for the aromatic dwarf variety. Note that SODCP candidate gene CN01_08G010630 (yellow star) is orthologous to AT2G28190 whose annotation is given in the box above the tree. See Supplementary Note 1 for results on the MT3 and GST genes (gray stars). **d, e** Same as (**a**, **b**) for the SOS pathway. **f** Same as (**c**) for the Shaker gene family. Note that AKT2 candidate gene CN01_01G020060 (yellow star) is orthologous to AT4G22200 whose annotation is given in the box above the tree.

(Supplementary Data 6–8). Supplementary Data 7 lists 152 coconut genes which are ortholog to genes known to be involved in salt-stress response in other plant species[7,9,71–73] and are assigned to five signaling pathways (ABA, ABAi, Ca$^{2+}$/PLC, ROS, and SOS). Finally, 65 genes were common to both sets (Supplementary Data 8). These signaling pathways are summarized in Table 2 and Fig. 6 and Supplementary Fig. S8, and the evolution of signaling gene expression during salt stress in depicted in Supplementary Fig. S9. More details are provided in Supplementary Figs. S10 and S11 for MT3 and ABA and ABAi families. Here, we focus on three key players.

Regarding the osmotic stress response, a gene on chromosome 8 coding for a chloroplastic Cu-Zn superoxide dismutase (SODCP) of the reactive oxygen species pathway (ROS; Supplementary Fig. S8a–c) was upregulated at 4 h in the tall but not in the dwarf (Fig. 6a–c), suggesting the SODCP expression level could be a marker of salt tolerance. In fact, oxidative stress signaling and ROS detoxification are both essential components of the underlying mechanisms[74]. This corroborates that the first reaction of a plant to salinity occurs within seconds to hours[9].

Expression of a PP2C gene on chromosome 14 (protein phosphatase 2C) of the abscisic acid (ABA) dependent pathway (Supplementary Fig. S11), was downregulated at all stages, contrary to what is was observed in rice and *Arabidopsis*[75]. This is however consistent with the fact that the physiological response to salt in dwarf coconut seedlings is mainly mediated by stomatal regulation[76]. In fact, a decrease of PP2C could enhance ABA signaling which, in turn, triggers stomatal closure as well as the synthesis of protective molecules contributing to the osmotic homeostasis[7].

Regarding ionic stress, we did not observe differential expression of overly salt-sensitive genes as in rice[7]. Instead, we noted a weak K$^+$-selective inward-rectifying channel (KIRC), which could be a candidate for sodium gateway and sensor with a role in cation homeostasis (SOS pathway; Fig. 6d, e). In fact, sodium may, under some circumstances, utilize potassium channels[77,78]. Moreover, the polypeptide presents a DIRFSY motif, reminiscent to the Na$^+$-sensitive motif DxRxxH[79]. This gene on chromosome 1 is orthologous to AKT2 in *Arabidopsis*[80,81] expressed thorough the entire plant[82] (see Fig. 6f). A protective effect of AKT2 against ionic shocks has been proposed for *Arabidopsis* roots[15] and this could apply in coconut leaves as well.

## Conclusions

We present here a chromosome-scale assembled coconut genome, which will serve as a reference in subsequent coconut genomic research and breeding programs. It paves the way to studying genomic diversity within a coconut, throwing light on the differences between tall and dwarf coconuts, as illustrated by the CATD sequence[26] and between the two major genetic groups of coconuts, Pacific and Indo-Atlantic[40]. Combining genomic and phenotypic data will be useful to identify putative genes involved in a QTL as well through a candidate gene approach. The present assembly is an important milestone toward a comprehensive representation of the coconut genome. While the gene-rich portion of the chromosomes (euchromatin) is faithfully represented, low recombination rates in the repeat-rich portion (heterochromatin) prevented its complete and accurate assembly. Methods such as Hi-C and optical mapping will be needed to enrich it.

We demonstrated a close synteny between coconut and oil palm, exhibiting 13 large synteny blocks covering most of their genomes. Our results confirm the palm evolution scenario

proposed in Murat et al.[49] in its broad lines while enriching it. Further paleogenomics research involving more species such as date palm is expected to complete the reconstruction of the ancestral genome of palms. Our results can also be exploited in breeding and in ecology; both resemblances and differences are of interest. While overall synteny allows results obtained in one species to be transferred to the other, differential gene amplification may help to pinpoint the evolutionary strategy involved in adaptation to different environments. We expect such an approach to be useful in further research on the mechanism of salt resistance in coconut and other crops. The results of a preliminary experiment suggest that coconut tree responds to salt stress by mobilizing a number of signaling pathways to trigger metabolic functions ranging from the elimination of reactive oxygen species to the Na$^+$/K$^+$ homeostasis and the closure of stomata in response to increased osmotic pressure.

Finally, we spotted a recent invasion of the coconut genome by transposable elements, which may point toward a dramatic reduction of the coconut population size due to the fluctuation of ocean levels at the beginning and at the end of the Pleistocene glaciations.

## Methods

### Construction of a high definition linkage map

*Mapping population*. A mapping population was produced at CNRA (Côte d'Ivoire). It is a BC$_1$ using the Malayan Yellow Dwarf (MYD) as a recurrent parent. The other parent was an elite palm (P994) of the West African Tall (WAT), chosen for its good combining ability with the MYD[83]. The F$_1$ hybrid between P994 (male parent) and MYD palms was planted in 1993 and a single progeny (115 02 24) was used as a pollen donor to produce the backcross. The MYD is self-pollinating and can be treated as a pure line[84], thus multiple individuals of this cultivar could be used as mother palms in order to speed up the process. Pollination was conducted under an isolation bag from April to October 2013. Seed nuts were collected, identified by a unique number, and raised in a polybag nursery before planting in October 2016.

*Genotyping*. Total genomic DNA extractions were performed manually by grinding fresh leaves using liquid nitrogen and MATAB[85] method. DNA samples were quantified with a Fluoroskan Ascent FL fluorometer (Thermo Fisher Scientific, Waltham, MA, USA). Genomic DNA quality was checked using agarose gel electrophoresis. Out of 320 progenies, 292 passed a legitimacy test based on 15 SSR markers (CnCirH7, CnCirE10, CnCirB12, CnCirC7, CnCirF2, CnCir C5, CnCirE2, CnCirE12, CnCirB6, CnCirA9, CnCirA3, CnCir H11, CNZ40, CnCirG11, CnCirC12) whose characteristics can be found in Tropgene DB[86]. A genomic library was prepared using PstI-MseI (New England Biolabs, Hitchin, UK) restriction enzymes with a normalized 200 ng quantity of DNA per sample. The procedures published by Elshire et al.[87] were followed; however, the common adapter was replaced to be complementary to MseI recognition site. Digestion and ligation reactions were conducted in the same plate. Digestion was conducted at 37 °C for 2 h and then at 65 °C for 20 min to inactivate the enzymes. The ligation reaction was done using T4 DNA ligase enzymes (New England Biolabs, Hitchin, UK) at 22 °C for 1 h, and the ligase was then inactivated by heating at 65 °C for 20 min. Ligated samples were pooled and PCR-amplified (18 cycles). The PCR-amplified libraries were purified using the Wizard PCR preps DNA purification system Promega (Madison, USA) and verified with the Agilent D5000 ScreenTape (Santa Clara, USA).

Single-end sequencing of 150 base-pair reads was performed in a single lane at the GeT-PlaGe platform in Toulouse, France in several batches. The first one in January 2016 (single reads of 100 bp on Illumina Hiseq 2500), two others in May 2016, this time yielding single reads of 150 bp on Illumina Hiseq 3000. Parents were replicated to ensure SNP detection and high-quality parental information for the estimation of marker segregation types. The MYD parent was included as a control in all batches, while P994 hybrid parent were only present in the batches sequenced in May. In total, they were repeated six and ten times, respectively. Based on sequencing statistics, 240 progenies were retained for bioinformatics analysis. Five fastq files were produced from the three batches (6 plates in total) and analyzed with TASSEL-GBS[33] pipeline V5.2.29 (see Supplementary Fig. S12 and Supplementary Table S1) on the Cirad HPC data center of the South Green Bioinformatics platform resulting in 631,283 SNP variants assigned to 7903 scaffolds (VCF file 9× read depth).

In an effort to enrich the map with large scaffolds, the allele calling was repeated, targeting only 368 of the largest scaffolds as yet unmarked and using the Process-Reseq pipeline. This time, we used the following programs: demultiplex.py program available at https://github.com/timflutre/quantgen; arcad_hts_1_cutadapt_in_chain.pl and

arcad_hts_2_Filter_Fastq_On_Mean_Quality.pl, which are part of Arcad-hts[88], available at https://github.com/SouthGreenPlatform/arcad-hts; and finally process_reseq.1.0.py pipeline and VcfPreFilter.1.0.py program. which are part of the VcfHunter[89] package, available at https://github.com/SouthGreenPlatform/VcfHunter. The parameters and summary outcome are in Supplementary Table S2.

This additional procedure initially identified 8,846,296 variants, of which 71,677 SNPs on 365 scaffolds were selected.

The first filter with a binomial test was applied (VcfFilter.1.0.py), leading to 17,548 SNP markers assigned to 3017 scaffolds for TASSEL-GBS (71,677 SNPs on 365 scaffolds for VcfHunter, see Supplementary Table S3). The parameters were:

- Minimal read number of minor allele to call variant heterozygous (1).
- Criteria to score data points: if the proportion of the minor allele is below 0.01, the locus is scored as homozygous. If it is above 0.1, it is scored as heterozygous. In between, missing data are generated.
- P-value threshold to keep a marker (1e-20).
- Maximal missing data proportion per marker (0.5).

In the process of constructing the data matrix, a further filtering process was conducted using a R[90] markdown procedure. The Tassel_to_J pipeline R and its tutorial are available at https://github.com/SouthGreenPlatform/Curation-GBS-data. The criteria were missing data <25% per individual and <20% per marker (see Supplementary Table S4). The procedure further discarded markers based on redundancy, abnormal segregations, and discordant segregation between markers of the same scaffold (outliers), and performed imputation and genotyping error correction at the scaffold level). After a final manual curation, the number of useful markers was reduced to 8402 SNPs, located on 2303 scaffolds (79 of which resulted from the Process-Reseq procedure. They belong to 49 large scaffolds totaling 41 Mb). In the process, the number of useful individuals was reduced to 216.

*Map construction and scaffold anchoring and enrichment.* Genotypes were coded as follows: "a" for the MYD parent, "h" for the heterozygote, and "u" for missing data, and the resulting genotyping matrix was imported into JoinMap 4.1 software to compute the recombination rate and LOD score and to identify linkage groups. JoinMap is efficient software to group individual loci into linkage groups and construct a linkage map, however, it is unable to handle scaffolds on which the order of markers is known a priori. For this reason, scaffolds assigned to each linkage group were ordered using Scaffhunter[36] using LOD scores on the HPC data center in order to build the AGP and fasta files of the pseudomolecules (see Supplementary Table S5).

*Computing linkage distance in presence of genotyping errors.* A well-known problem with GBS maps results from the accumulation of genotyping errors, which result in grossly overestimated recombination distances. Even if this rate is low (<1% in our case) they are cumulated over the numerous loci obtained with this technique, resulting in unrealistic linkage distances. To alleviate this problem, we applied the same imputation and genotyping error correction procedure as above, but at the linkage group level and calculated linkage distances based on corrected data. Cirad R markdown code is available at https://github.com/SouthGreenPlatform/Curation-GBS-data.

*Recombination, protein gene, and marker landscapes.* Plots on Supplementary Fig. S4 were produced with an R markdown procedure. For the recombination landscape, we used a table with the positions of the markers on the chromosome (in base pairs) and the distance to the previous market in cM). The latter is converted into a stair-step function using the "cumsum" function. This function is smoothed and estimated at 1 Mb intervals using "loess" and "predict" functions. The result is differentiated numerically: $(f(x + 1) − f(x − 1)/2)$ where $x$ is the position in Mb and $f$ the smoothed function. For the first and the last marker of a chromosome, the derivatives become $f(1) − f(0)$ and $f(x_{max}) − f(x_{max} − 1)$, respectively.

The same procedure is applied for the marker landscape except that the distances between markers are replaced by 1. For protein genes, the positions of the genes are used instead of the marker position.

*Locating locus under selection and estimating selection pressure.* A strong selection in favor of homozygous genotypes was detected in chromosome Cn15. In order to locate the selected gene, we modeled the segregation disequilibrium as a linear function of the linkage distance between a given locus and the locus under selection. Specifically, the expected frequency of heterozygotes at a locus ($F_{ab}$) is

$$F_{ab} = \frac{100 \times d}{e} \times \frac{1 - s}{1 + s}$$

where $d$ is the estimated distance (in cM) to the locus at which selection occurs; $s$ is the relative fitness of heterozygotes (the fitness of a homozygote being equal to 1 by convention), and $e$ is a correction factor taking into account possible discrepancies between the estimated and actual map lengths. These three parameters were jointly estimated using an iterative non-linear minimization procedure. We estimated graphically plausible values and improved them by minimizing the sum of squares of the differences between the observed and expected values of $F_{ab}$ over all loci using function nlm() of R language. Cirad R markdown code is available at https://github.com/SouthGreenPlatform/Curation-GBS-data.

*Estimated insertion times of LTR retrotransposons.* To estimate the insertion times for the full-length LTR retrotransposons in the *C. nucifera* and *Phoenix dactylifera* genomes, intact LTR retrotransposons (contained the 5′ and 3′ LTR sequences of the retrotransposons) were identified de novo using LTR_STRUC with default parameters. Then the 5′ and 3′ LTR sequences of retrotransposons were aligned using MUSCLE[91], and the distance $K$ between them (the average number of substitutions per aligned site) was calculated with Kimura two-parameter model using the distmat program implemented in EMBOSS v6.5.0 (see URLs). The insertion time $T$ was calculated as $T = K/(2 \times r)$, where r represents the rate of nucleotide substitution, which was set based on a mutation rate of $7 \times 10^{-9}$ per site per year.

This takes in account the substitution rate in palms estimated as $2.61 \times 10^{-9}$ for ADH[92] and the fact that TEs are known to evolve two to three times faster than genes[93].

*Ks analysis.* Pairs of paralogous genes were identified by filtering the BlastP alignments between coconut polypeptides annotated on scaffolds, allowing a minimum of 30% identity and a minimum of 40% coverage and are represented in Circos[94] (Fig. 1). Several best hits are allowed for each protein. The synonymous mutation rate (Ks) was calculated for each pair of paralogues using the paml software[95]. The distribution of the natural logarithm of the Ks was plotted. This distribution was analyzed with Emmix software[96]. We tested 1–10 components with Emmix software and repeated the EM algorithm 100 times with random starting values, and 10 times with k-mean start values. The decomposition in components with the lowest Bayesian information criterion value was selected for subsequent analysis. The distributional peaks suggested by Emmix were verified with SiZer software[97] according to the significant slope changes. Emmix was used again to obtain the number of peaks confirmed by SiZer, the groups of Ks were represented in a dot plot in R (Supplementary Fig. S5).

*Identifying pre-p-WGD ancestor.* Synteny blocks between the paralogous regions of coconut pseudochromosomes were determined with DRIMM-synteny software[98]. In total, 13 blocks of duplication were identified in the oil palm genome, we also identified these duplicate blocks using the orthology between these species in the coconut paralogous and syntenic regions. The ancestral chromosomes of the pre WGD p were reconstructed from these 13 blocks with MGRA software[50]. These ancestral chromosomes were verified by manual verification. The ancestral chromosomes are represented in a dot plot (Supplementary Fig. S6).

*Orthology between coconut and oil palm.* In parallel, the coconut polypeptides were blasted onto the *Elaeis guineensis* proteome of the NCBI Reference Sequence (RefSeq) database[99], annotation release 100, in order to exploit synteny based on coconut best BLASTP hit (BBH). The best hit for coconut proteins was identified as the alignment with the highest identity and coverage of coconut protein. These results were filtered to discard oil palm proteins exhibiting multiple homology relationships. Results are represented in the form of a dot plot where each dot corresponds to a pair of orthologous protein sequences (Supplementary Fig. S7).

*Signaling pathways involved in response to salinity stress.* A salt-stress experiment was conducted on coconut palms seedlings in a greenhouse at the Coconut Research Institute (CRI), Wenchang City, Hainan CHINA. Leaf samples were collected for transcriptome analysis. A sampling at time point 0 was used as a control and further samples were taken at 4 h, 6 days, and 10 days after NaCl treatment. The coconut seedling used in this study is about four to 5-months old with about 50 cm in height in a container located in the greenhouse. The seedlings were irrigated with 1/4 strength of Murashige and Skoog (MS) enriched with 200 mM NaCl (about a third of the salt concentration of seawater) weekly. The experiments were done in three replications (three seedlings), but only two samples per variety were available for 10 days.

Varieties used are Hainan tall (BD) which was been used for genome sequencing and aromatic dwarf (XS).

A sampling at 0 time point was used as a control, samples were taken at 4 h, 6 days, and 10 days after 200 mM NaCl treatment. The RNA samples were collected from three different leaves but one to two samples were missing for 10-days sampling.

The growth medium was kept wet during treatment. The A1, A2, and A3 were collected on different leaves (all the samples named A1 are dependent samples collected on the same leave).

*RNA extraction and sequencing.* RNA was isolated by modified CTAB methods. Briefly, about 60 mg of plant tissue was ground in liquid nitrogen into a fine powder and transferred into the tubes contained preheated CTAB buffer with 2% of beta-mercaptoethanol. After 20 min incubation at 65 °C, the mixture was centrifuged at $12,000 \times g$ for 5 min. The following procedure was applied: transfer upper phase into a new tube and for each ml of supernatant add 200 μL of chloroform:isoamyl alcohol (24:1, v/v), mix using vortex. after centrifugation at $12,000 \times g$ for 5 min, transfer the upper phase into a new 2.0-ml tube and add an equal volume of phenol:chloroform:Isoamyl alcohol (25:24:1, v/v), and mix by vortex. After centrifugation at $12,000 \times g$ for 5 min at 4 °C, the upper phase was

transferred into a new 2.0-ml tube and add an equal volume of chloroform:isoamyl alcohol (24:1, v/v), mix by vortex. Transfer upper phase into a new 1.5-ml tube after centrifuging at $12,000 \times g$ for 5 min at 4 °C, add equal volume isopropanol, and mix by inverting the tube up and down. After leaving the mixture at −20 °C for 1 h, centrifuge at $20,000 \times g$ for 20 min to precipitate the pellet and wash the pellet with 75% ethanol. Dry the pellet and dissolve the pellet in 30–50 μL of RNase-free water for the next application.

The total RNA samples were first treated with DNase I to degrade any possible DNA contamination. Then the mRNA was enriched by using the oligo(dT) magnetic beads. Mixed with the fragmentation buffer, the mRNA was fragmented into short fragments. Then the first strand of cDNA was synthesized by using a random hexamer-primer. Buffer, dNTPs, RNase H, and DNA polymerase I were added to synthesize the second strand. The double-strand cDNA was purified with magnetic beads. End reparation and 3′-end single nucleotide A (adenine) addition was then performed. Finally, sequencing adaptors were ligated to the fragments, and sizes from 100 to 500 bps were selected for library construction. The fragments were enriched by PCR amplification. During the QC step, Agilent 2100 Bioanaylzer and ABI StepOnePlus Real-Time PCR System were used to qualify and quantify the sample library. The library products were then sequenced on Illumina HiSeqTM 4000 with SE50.

*Processing of the RNA-Seq data.* RNA-Seq analyses were done following the workflow described in Supplementary Fig. S13. Primary sequencing data produced by Illumina HiSeqTM 4000 was subjected to quality control (QC). After QC, raw reads were filtered into clean reads which were aligned to the reference sequences. If the alignment result passed QC, the clean reads were processed with downstream analysis including gene expression and deep analysis based on gene expression (PCA/correlation/screening differentially expressed genes and so on). Clean reads were mapped to genome reference (scaffolds) using BWA[100] and to gene reference using Bowtie[101]. In this analysis, both alignment tools were used to assess transcriptome, genome, and gene annotation quality. Bowtie was also used as part of RSEM[102] (RNA-Seq by Expectation Maximization) gene expression quantification workflow.

Differentially expressed genes were screened using Noiseq method according to the following criteria, A gene was considered upregulated (resp. downregulated) at a given time (4 h, 6 days, or 10 days) if three conditions were simultaneously met:

- The mean level of expression (FPKM) at the considered time was at least (resp. at most) 100.
- This level was at least twice (resp. at most half) the level at the initial time.
- The probability of a significant difference is at least 0.8.

We considered several types of differentially Ea+ or Ea- labels were assigned to genes if they were upregulated or downregulated at 4 h. Likewise, received labels La + or La− if they were upregulated or downregulated at either 6 days or 10 days. These "early" or "late" regulation labels could be combined eg. Ea−La+. Finally, we considered a gene as constitutively highly expressed (labeled CHE) if its level of expression exceeded 350 and was not differentially expressed. Only differentially expressed and CHE genes are included in Supplementary Tables S3 and S5.

We distinguished nine classes of temporal expression patterns and assigned 66 expression classes numbers (ECN): genes having the same ECN have similar putative functions and expression profiles

The hierarchical clustering of the expression patterns was performed by R using the Z-scores of the normalized read counts and represented in a heatmap using gplots and RColorBrewer libraries. R markdown code (heatmap-cluster_02.rmd) is available at https://github.com/SouthGreenPlatform/Myrtaceae-RNASeq-scripts.

*Signaling pathway and gene family analysis.* We looked for coconut gene which is orthologous to genes involved in five signaling pathways reported in previous studies[7,9,71–73]: osmotic stress signaling transduced by the abscisic acid-dependent (ABA) or independent (ABAi) pathways, the reactive oxygen species (ROS) pathway, the ionic stress signaling based on $Ca^{2+}$/phospholipase C ($Ca^{2+}$/PLC) and on salt overly sensitive genes (SOS) pathway. Following previous studies, we treated the calmodulin (CaM) pathway, which exerts the same kind of regulation, together with SOS. In addition, not having found differential expression in the SOS pathway, we considered the Shaker family where AKT2 is differentially expressed.

The gene family analysis consisted in phylogenomic and synteny analyses of six genomes: coconut V1, RefSeq annotation release 101 for date palm (assembly GCF_000413155.1_DPV01), oil palm (GCF_000442705.1_EG5) and rice (GCF_001433935.1_IRGSP-1.0), banana V2[36], Arabidopsis TAIR10[103] using workflows "GenFam: Hmmer Multi Species", "GreenphylDB for proteins" and "GenFam: Visualisation" of the South Green Galaxy instance: http://galaxy.southgreen.fr/galaxy/workflow/list_published. The last one uses the InTreeGreat[104] visualization web interface integrating the synteny and differential expression results along a phylogenomic tree (Fig. 6 and Supplementary Figs. S10 and S11c, e). The preliminary syntenic analysis, based on GoGe Synmap[105] workflow, could predict the type of duplication such as whole-genome duplications (WGD). The differential expression was represented by red or blue squares (low or high expression respectively) (downregulated gene) and blue (upregulated gene) squares disposed of in six columns: (1) tall 4 h, (2) tall 6 days, (3) tall 10 days, (4) dwarf 4 h, (5) dwarf 6 days, (6) dwarf 10 days.

**Reporting summary**. Further information on research design is available in the Nature Research Reporting Summary linked to this article.

## Data availability
The final assembly and annotation are deposited in DDBJ/EMBL/GenBank under the following identifiers: SUBID: SUB5865736, BioProject: PRJNA374600, BioSample: SAMN06328965, Accession: VOII00000000, Organism *Cocos nucifera*. The genetic map, mapping population genotypes, genome sequence, and annotation can be obtained and viewed at http://palm-genome-hub.southgreen.fr/. Transcriptomic data (RNA-Seq and differential expression) are available under GEO Superseries Accession GSE134410. Mapping population GBS data are available under BioSample accessions: SAMN15659886 to SAMN15660159 and SRA runs 15085388 to 15085437.

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

## Acknowledgements

This study was supported by the fundamental Scientific Research Funds for Chinese Academy of Tropical Agriculture Sciences (CATAS-Nos. 1630032012044, 1630052014002, 1630052015050, and 1630152017019, Central Public-interest Scientific Institution Basal Research Fund for Innovative Research Team Program of CATAS (NO. 17CXTD-28), the International Science and Technology Cooperation projects of Hainan Province (No. KJHZ2014-24), the major Technology Project of Hainan (No. ZDZX2013023-1), the National Natural Science Foundation of China (No. 32071805). The production of the mapping population received funding from COGENT and Bioversity international through projects LOA C Coconut 2012/45 and 2014/17. Genotyping and comparative genomics analysis were supported by Montpellier Supagro fund. Guillaume Martin and Gaetan Droc provided helpful advice for mapping and for the construction of the Palm Comparomics genome hub. The authors acknowledge the CIRAD UMR-AGAP HPC (South Green Platform) at CIRAD Montpellier for providing HPC resources that have contributed to the research results reported within this paper. http://www.southgreen.fr. They thank heartily Dr. D. Lantican and his co-authors for kindly providing us the annotation data of the CATD sequence.

## Author contributions

Y.Y., H.F., Y.X., W.X., M.P., A.G., L.B., and D.T. designed the study and contributed to project coordination. H.F., A.I., R.Q., L.Z., J.L., R.R., and Y.W. prepared samples and performed experiments. S.B., A.A., L.B., P.X., J.X., D.T., N.L., W.X., and Y.X. analyzed the data. Z.M. was responsible for coconut seedling management. A.E.I. produced the mapping population and collected samples for DNA analysis. C.H. was responsible for data management. Y.Y., B.S., L.B., and A.A. wrote the article.

## Competing interests

The authors declare no competing interests.
