## [Peer Review File · Communications Biology]

Reviewers' comments:

Reviewer #1 (Remarks to the Author):

Summary of the key results

=====

The manuscript titled "A reference sequence of coconut genome highlights major evolutionary steps among palms" describes the development of a coconut genetic map, the anchoring of the 47% of a previously published draft genome to 16 linkage groups. The manuscript also describes a repetitive element landscape analysis of this genome with three interesting observations: 1- TE accumulates in some of the chromosome centromeres; 2- TE accumulates in regions where potentially two ancestral chromosomes were fused; 3- Coconut LTR elements were expanded 0.5-1 MYA; Finally the manuscript presents the results of a transcriptomic analysis of two coconut varieties, Tall and Dwarf under salt stress. 590 genes were differentially expressed, of which the authors discuss three of them (SODCP, PP2C and KIRC) as particularly relevant for the response to salt stress.

Overall evaluation

=====

The manuscript present serious defects in its structure, making difficult the follow across the different sections. Some of the them are not well integrated with the rest of the text, such as the salt stress experiment. In terms of scientific support, the authors do not present any quality control in the development of the genetic map. They do not incorporate either a systematic comparison with previous works such as other genetic maps or other genomes. They imply in several part of the text that "the other genome" quality is lower without any supporting data. The analysis of the evolution of the karyotypes is interesting. It is difficult to evaluate if the RNASeq analysis was correctly performed through the lack of details in the Material and Methods, but some of the tools used (e.g. read mapping tools) were not designed to align RNASeq reads. In the same context, material and methods lacks in general of important details including the version of the program used. Finally, I do not think that the results of the manuscript as they are currently presented justify a publication in Communication Biology by themselves. No quality control, no biological story, no cohesion between different parts where some of them look unfinished (e.g. low percentage of the anchored genome, no comparison with other works...). I think that they may be interesting and relevant for the community, but I also think that they have not been fully developed for a full publication.

Major concerns

=====

- No real improvement in the genome assembly compared with the previous published coconut genome. The authors published the draft coconut genome in 2017 (Xiao et al. 2017, Gigascience). No extra sequencing using long read sequencing technologies, or new scaffolding methodologies such as optical mapping or HiC were incorporated in this manuscript. I recognize that the incorporation of a genetic map for anchor the draft to pseudomolecules is useful, but the authors could anchor only 47% of the draft looking like an unfinished job rather than a real improvement.
- No independent evaluation of the reliability of the genetic map used in the anchoring. There is not any analysis or experiment to give some idea of how reliable is the anchoring of the sequences to a genetic map. Some options could involve the comparison with already published genetic maps (e.g. Herran et al. 2000; Lebrun et al. 2001) or with newly developed ones, or the use of FISH experiment to visualize some of the genetic features of the coconut genome (Santana Pereira et al. 2017), or HiC (Latican et al. 2019).
- Absence of a systematic comparison of this genome assembly with previous ones. The authors present a "new" genome version in which they have anchored a previous draft to linkage groups. Nevertheless, a new genome draft was published recently (Latican et al. 2019). How is the genome

described in this manuscript compared with this one? The Latigan's genome draft has 34,958 gene models meanwhile the genome presented in this manuscript has only 28,039. The BUSCO values published previously (Xiao et al. 2017, Latican et al. 2019) indicates that this version has captured a lower percentage of the gene space than the Latigan's version. The authors should do a systematic comparison of their version with this one in order to highlight the "novelty" or "differences" between the two versions and so far, guide the scientific community in the use of the different version.

- Poorly structured manuscript/Lack of biological questions/Unprecise language: The structure of the manuscript is chaotic and it makes difficult to follow the story. It starts with the genetic map, and then continues with the segregation distortion analysis in which the authors bring to the table a discussion about the pollen of one of the parents of the population without previous description of this concept in the introduction. Then, the manuscript describe the anchoring of the sequence draft to the linkage groups, but it describe gene and repeat abundance without describe the origin of these elements from previous work. The WGD section, fits well after the previous one, but then jumps into the salt stress experiment. Related with the structure, the authors named the section "Results" but it is more "Results and Discussion" and "Discussion" should be "Conclusions". On top of that, the manuscript lack of a biological story. I agree that new genomes are important resources, but the authors should build the manuscript with a biological story and some hypothesis. Why the authors made the salt stress experiment? How is this connected with the genetic map? Finally, the authors should polish the scientific language used in the manuscript. For example, they wrote in the introduction "The quality of this sequence appears to be appreciable". Quality is a relative term in this context, is it good or bad, high or low?. Other example is where they wrote "By aligning 28,039 coconut gene models among themselves". Gene models are coordinates in a sequence that indicates where different features of the gene (e.g. exon) starts and ends. What you align, are sequences, and in this case for the Ks analysis, CDS sequences obtained from the gene models. They also use the term "paper" that it is quite informal, instead "publication" or "article".
- Lack of details in the Material and Method section. Please see minor concerns for details.
- RNA-Seq analysis. It presents two major problems. First, the lack of details in the material and methods makes difficult to evaluate if it was performed adequately (e.g. number of individuals per biological replicate). Second, the tools used for align the reads were not design for RNA-Seq but for DNA-Seq. Authors should use tools like Tophat, Hisat2, Star... Third, no information about number of reads, mapping percentage... was supplied.

Minor concerns

=====

- Title: The title drives to a miss-interpretation of the content of the manuscript. For the title, the reader could assume that the authors are presenting a new genome, but it is indeed an "update" based in a genetic map.
- Introduction, page 3, line 16: In the introduction, the authors wrote "The one used for CATD appear to be efficient for assembling gene rich sequences, while missing part of the transposable elements (TE)". Do they have support of this affirmation? Usually long read assemblies make a much better job assembling TE than short read based assemblies.
- Material and Method. The description about the development of the population is incomplete. Details about the direction of the cross (who were maternal/paternal progenitors in each cross) are missing.
- Material and Methods: How was extracted the DNA used in the genotyping? Please describe step by step the methodology and material used for the DNA extraction and quality evaluation.
- Material and Methods: Which protocol was used for the GBS libraries? Please cite. Where the GBS were sequenced? Was TASSEL-GBS run with the default parameters?
- Material and Methods: Details about the construction of the genetic map are missing. E.g. distance method used, LOD score, min. RFs...
- Material and Methods: The version of the program used for all the analysis is missing. Please

complete, and add "with the default parameters" if it was run with them.

- Material and Methods: Please detail the version of the other genomes used in the analysis (and if it is possible, the AccessionID from the database that it was downloaded).
- Material and Methods: For the salt stress experiment, the authors did not specify how many individuals were used per independent biological replicate.
- Material and Methods: The authors did not specify if size selection was performed over the RNA-Seq library and which average size was selected. They did not detail also if it was sequenced single or pair ends, and the length of the read, and where it was sequenced.
- Material and Methods: BWA or Bowtie are not commonly used to align RNA-Seq reads to a reference genome because they are not splice aware. Usually tools like Tophat, Hisat2 or Star are used instead.

Reviewer #2 (Remarks to the Author):

Remarks to the Author:

The manuscripts presents a chromosome level assembly of the coconut genome scaffolded with linkage maps using GBS. The paper is well structured with the appropriate literature review, analysis and conclusions.

Summary:

The manuscript presents a chromosome level assembly of the coconut genome scaffolded with a linkage map using GBS. The authors further characterise the evolution of this genome highlighting close synteny with date palms.

Inference is made on the possible ancestral genome of the coconut based on ancestral genome reconstruction, and the possible role of transposable elements in the evolution of coconuts. An estimate of the time of TE insertions and whole genome duplication events are also provided. Overall, the paper is well written, with analytical rigour. Results presented will be of great interest to coconut researchers and plant biologists in general.

Comments:

The claims of the paper are convincing. The additional analysis suggested would be of great importance, as noted below.

Lines 171-180: The authors claim that based on patterns of the distribution of TE's on chromosomes Cn09, Cn12, Cn13 and Cn16, that these chromosomes may be acrocentric. The distribution observed could have been as a result of incomplete chromosome scaffolds that were not placed by the TASSEL pipeline. Did the authors do any checks on sequences not anchored? The distribution of TE's on these may indicate ends of parts of chromosomes Cn09, Cn12, Cn13 or Cn16.

Lines 287 -288: The authors in the methods state that they collected RNA from leaf samples to identify genes expressed under high salinity. Did the authors consider doing the same analysis on root tissues?

Lines 340-341: The authors further state that they did not observe overly sensitive expression of genes to salt stress as is the case reported in rice. Would this be as a result of doing a differential

expression analysis of leaf samples only and not root? Papers of interest here include (Moons A, Prinsen E, Bauw G, Van Montagu M: Antagonistic effects of abscisic acid and jasmonates on salt stress-inducible transcripts in rice roots. Plant Cell 1997, 9: 2243–2259).

Lines 368-369: The authors on estimating the insertion time of LRTs highlight that they used a 50% lower substitution rate as that reported in the CATD paper. Would the authors expand their choice of a lower substitution rate despite the conclusions of their results being similar to the CATD paper or why they believe that the earlier substitution rate used in the CATD paper was high hence their choice of a lower rate?

Methods:

Lines 587 : Needs a re-write for clarity.

Lines 594-610: The differential gene expression screening using Noiseq method would be better presented as a flow diagram.

Kind regards,

Paul

Dr Paul Visendi Muhindira

Data availability for reviewers

Data are under restricted access until publication. However, editors and reviewers will be able to access them:

The chromosome assembly and accompanying data were submitted to NCBI and received the following identifiers: SUBID: SUB5865736, BioProject: PRJNA374600, BioSample: SAMN06328965, Accession: VOII00000000, Organism *Cocos nucifera*.

GBS data read data are now available under SRA accessions SAMN15659886 to SAMN15660159.

These data can be accessed at the palm.comparomics site (see below).

- RNA-Seq data are readily accessible on the GEO portal as GSE134410 using token mxytawuqfzbpml.
- In addition, these data and various exploration tools are accessible at “palm.comparomics” (<http://palm-genome-hub.southgreen.fr/>). Login as reviewer with password COCNU2019 and answer “easy math question”). Downloadable files are accessible at <http://palm-genome-hub.southgreen.fr/downloadable-files-cocos-nucifera-studies>.
- in house procedures and documentation available at <https://github.com/SouthGreenPlatform/Curation-GBS-data>.

Response to reviewers' comments:

We would firstly thank the reviewers for their detailed and useful remarks and hope that the new version will meet their expectations. Reviewer 1 proposed to use Hi-C and optical mapping to assemble the genome. While we agree that it would considerably improve the assembly (and will in a future version), the tests we made show that Hi-C cannot be used alone. In other words, what needs to be done is to combine linkage mapping and the other methods, a rather complex task and this would delay considerably the publication of a genomic resource that is much needed by the coconut genetic improvement community. The admittedly large missing portion (53%) of the genome represents only 23% of the genes.

Reviewer #1

Summary of the key results

The manuscript titled “A reference sequence of coconut genome highlights major evolutionary steps among palms” describes the development of a coconut genetic map, the anchoring of the 47% of a previously published draft genome to 16 linkage groups. The manuscript also describes a repetitive element landscape analysis of this genome with three interesting observations: 1- TE accumulates in

some of the chromosome centromeres; 2- TE accumulates in regions where potentially two ancestral chromosomes were fused; 3- Coconut LTR elements were expanded 0.5-1 MYA; Finally the manuscript presents the results of a transcriptomic analysis of two coconut varieties, Tall and Dwarf under salt stress. 590 genes were differentially expressed, of which the authors discuss three of them (SODCP, PP2C and KIRC) as particularly relevant for the response to salt stress.

Overall evaluation

The manuscript present serious defects in its structure, making difficult the follow across the different sections. Some of the them are not well integrated with the rest of the text, such as the salt stress experiment. In terms of scientific support, the authors do not present any quality control in the development of the genetic map. They do not incorporate either a systematic comparison with previous works such as other genetic maps or other genomes. They imply in several part of the text that “the other genome” quality is lower without any supporting data. The analysis of the evolution of the karyotypes is interesting. It is difficult to evaluate if the RNASeq analysis was correctly performed through the lack of details in the Material and Methods, but some of the tools used (e.g. read mapping tools) were not designed to align RNASeq reads. In the same context, material and methods lacks in general of important details including the version of the program used. Finally, I do not think that the results of the manuscript as they are currently presented justify a publication in Communication Biology by themselves. No quality control, no biological story, no cohesion between different parts where some of them look unfinished (e.g. low percentage of the anchored genome, no comparison with other works...). I think that they may be interesting and relevant for the community, but I also think that they have not been fully developed for a full publication.

➤ **See answers below**

Major concerns

- No real improvement in the genome assembly compared with the previous published coconut genome. The authors published the draft coconut genome in 2017 (Xiao et al. 2017, Gigascience). No extra sequencing using long read sequencing technologies, or new scaffolding methodologies such as optical mapping or HiC were incorporated in this manuscript. I recognize that the incorporation of a genetic map for anchor the draft to pseudomolecules is useful, but the authors could anchor only 47% of the draft looking like an unfinished job rather than a real improvement.

➤ **It is true that the present takes the draft assembly as is. You describe this assembly as an “unfinished job” and we feel that it depends on which part of the genome is considered. What we show in the revised version is that the distribution of markers and of crossing overs indicates that euchromatin is likely almost completely and accurately assembled while heterochromatin (which represent a large portion of the coconut genome) is incompletely and rather imprecisely assembled. This was confirmed by a preliminary comparison with Hi-C results (not shown). Hi-C, along with optical mapping (in progress) will be used for a future revised version, allowing a much more complete heterochromatin assembly. As suggested in the new version, linkage mapping data will still play a crucial role.**

- No independent evaluation of the reliability of the genetic map used in the anchoring. There is not any analysis or experiment to give some idea of how reliable is the anchoring of the sequences to a genetic map. Some options could involve the comparison with already published genetic maps (e.g. Herran et al. 2000; Lebrun et al. 2001) or with newly developed ones, or the use of FISH experiment to visualize some of the genetic features of the coconut genome (Santana Pereira et al. 2017), or HiC (Latican et al. 2019).

- **We included a more complete assessment of the linkage map and of the resulting assembly, in addition to figure S1, which shows that the mapping results are consistent, figure S2 shows that Hi-C is liable to improve many small imperfections due to linkage mapping but occasionally introduces large errors (ie. chimeric assemblies). Figure S4 shows that the marker density and the recombination rate are sufficient to ensure a fairly accurate assembly in euchromatin.**
- **We chose not to include a comparison with the previous map by Lebrun et al. 2001, completed in Baudouin et al. 2006 because it would rely on only 42 SSR. This comparison was made anyway and only one serious discordance was found, where the previous map had a 30 cM gap between markers.**

- Absence of a systematic comparison of this genome assembly with previous ones. The authors present a “new” genome version in which they have anchored a previous draft to linkage groups. Nevertheless, a new genome draft was published recently (Latican et al. 2019). How is the genome described in this manuscript compared with this one? The Latican’s genome draft has 34,958 gene models meanwhile the genome presented in this manuscript has only 28,039. The BUSCO values published previously (Xiao et al. 2017, Latican et al. 2019) indicates that this version has captured a lower percentage of the gene space than the Latican’s version. The authors should do a systematic comparison of their version with this one in order to highlight the “novelty” or “differences” between the two versions and so for, guide the scientific community in the use of the different version.

- **We amplified the comparison. The BUSCO analysis suggests that the CATD has a little more duplicated genes while a little more functions are missing in the HAIT. The distributions of gene lengths in both genome assemblies are compared in figure S3. Other pending studies on coconut and comparison with related palms suggest that the actual gene number would be somewhere between the two estimates, perhaps around 30,000 genes.**

- Poorly structured manuscript/Lack of biological questions/Unprecise language: The structure of the manuscript is chaotic and it makes difficult to follow the story. It starts with the genetic map, and then continues with the segregation distortion analysis in which the authors bring to the table a discussion about the pollen of one of the parents of the population without previous description of this concept in the introduction. Then, the manuscript describe the anchoring of the sequence draft to the linkage groups, but it describe gene and repeat abundance without describe the origin of these elements from previous work. The WGD section, fits well after the previous one, but then jumps into the salt stress experiment.

- **This was certainly a shortcoming of the previous version. We present here a justification of the salt study in the introduction and at the beginning of this section.**

Related with the structure, the authors named the section “Results” but it is more “Results and Discussion” and “Discussion” should be “Conclusions”.

- **Regarding the section title, we followed the recommendations of Nature Communications. If ever this should be changed for NCB, this could be done easily.**

On top of that, the manuscript lack of a biological story. I agree that new genomes are important resources, but the authors should build the manuscript with a biological story and some hypothesis. Why the authors made the salt stress experiment? How is this connected with the genetic map?

- **See above. The current study is actually a follow up to the study of the ion transporter families made in the previous paper.**

Finally, the authors should polish the scientific language used in the manuscript. For example, they wrote in the introduction “The quality of this sequence appears to be appreciable”. Quality is a relative term in this context, is it good or bad, high or low? Other example is where they wrote “By aligning 28,039 coconut gene models among themselves”. Gene models are coordinates in a sequence that indicates where different features of the gene (e.g. exon) starts and ends. What you align, are sequences, and in this case for the Ks analysis, CDS sequences obtained from the gene models. They also use the term “paper” that it is quite informal, instead “publication” or “article”.

- **We did our best to correct these imperfections.**

- Lack of details in the Material and Method section. Please see minor concerns for details.
- RNA-Seq analysis. It presents two major problems. First, the lack of details in the material and methods makes difficult to evaluate if it was performed adequately (e.g. number of individuals per biological replicate). Second, the tools used for align the reads were not design for RNA-Seq but for DNA-Seq. Authors should use tools like Tophat, Hisat2, Star... Third, no information about number of reads, mapping percentage... was supplied.

- **We included the required information in the Material and Methods section, in supplementary data 3 and in sections 6 and 7 of supplementary information.**

Minor concerns

- Title: The title drives to a miss-interpretation of the content of the manuscript. For the title, the reader could assume that the authors are presenting a new genome, but it is indeed an “update” based in a genetic map.

- **The title was amended accordingly**

- Introduction, page 3, line 16: In the introduction, the authors wrote “The one used for CATD appear to be efficient for assembling gene rich sequences, while missing part of the transposable elements (TE)”. Do they have support of this affirmation? Usually long read assemblies make a much better job assembling TE than short read based assemblies.

- **Our claim was based on a comparison between the overall composition of the genomes and on the results of flow cytometry estimation. We don’t have a precise explanation for the difference.**

- Material and Method. The description about the development of the population is incomplete. Details about the direction of the cross (who were maternal/paternal progenitors in each cross) are missing.

- **Done. The Dwarf is the female parent in both generations.**

- Material and Methods: How was extracted the DNA used in the genotyping? Please describe step by step the methodology and material used for the DNA extraction and quality evaluation.

- **The information is given in the new version.**

- Material and Methods: Which protocol was used for the GBS libraries? Please cite. Where the GBS were sequenced? Was TASSEL-GBS run with the default parameters?

- **The information is given in the new version.**

- Material and Methods: Details about the construction of the genetic map are missing. E.g. distance method used, LOD score, min. RFs

- **Done. LOD score was used.**

- Material and Methods: The version of the program used for all the analysis is missing. Please complete, and add “with the default parameters” if it was run with them.

- **Done, see section 6 and 7 of Supplementary information for details.**

- Material and Methods: Please detail the version of the other genomes used in the analysis (and if it is possible, the AccessionID from the database that it was downloaded).

- **Done.**

- Material and Methods: For the salt stress experiment, the authors did not specify how many individuals were used per independent biological replicate.

- **Done (three individuals).**

- Material and Methods: The authors did not specify if size selection was performed over the RNA-Seq library and which average size was selected. They did not detail also if it was sequenced single or pair ends, and the length of the read, and where it was sequenced.

- **Done?**

- Material and Methods: BWA or Bowtie are not commonly used to align RNA-Seq reads to a reference genome because they are not splice aware. Usually tools like Tophat, Hisat2 or Star are used instead.

- **Bowtie was used here because it is the recommended alignment tool in the RSEM workflow.**

Reviewer #2

Remarks to the Author:

The manuscript presents a chromosome level assembly of the coconut genome scaffolded with linkage maps using GBS. The paper is well structured with the appropriate literature review, analysis and conclusions.

Summary:

The manuscript presents a chromosome level assembly of the coconut genome scaffolded with a linkage map using GBS. The authors further characterise the evolution of this genome highlighting close synteny with date palms.

Inference is made on the possible ancestral genome of the coconut based on ancestral genome reconstruction, and the possible role of transposable elements in the evolution of coconuts. An estimate of the time of TE insertions and whole genome duplication events are also provided. Overall, the paper is well written, with analytical rigour. Results presented will be of great interest to coconut researchers and plant biologists in general.

Comments:

The claims of the paper are convincing. The additional analysis suggested would be of great importance, as noted below.

Lines 171-180: The authors claim that based on patterns of the distribution of TE's on chromosomes Cn09, Cn12, Cn13 and Cn16, that these chromosomes may be acrocentric. The distribution observed could have been as a result of incomplete chromosome scaffolds that were not placed by the TASSEL pipeline. Did the authors do any checks on sequences not anchored? The distribution of TE's on these may indicate ends of parts of chromosomes Cn09, Cn12, Cn13 or Cn16.

- **It is theoretically possible, but we made a comparison with an assembly of the same genome using Hi-C and it does not seem to be the case. These results are not shown but will be used for a version 2 of the sequence.**

Lines 287 -288: The authors in the methods state that they collected RNA from leaf samples to identify genes expressed under high salinity. Did the authors consider doing the same analysis on root tissues?

- **It's absolutely true that a similar analysis should be done on the roots. However, this would have required a much more complex experimental protocol for this preliminary study. Coconut growth is slow and relatively young plants in polybags were used and collecting roots would probably have disturbed the plant growth. Without appropriate precautions, this could have interfered seriously with salt stress response. A study of the leaves gives useful information eg. about stomatal closure mediated by ABA.**

Lines 340-341: The authors further state that they did not observe overly sensitive expression of genes to salt stress as is the case reported in rice. Would this be as a result of doing a differential expression analysis of leaf samples only and not root? Papers of interest here include (Moons A, Prinsen E, Bauw G, Van Montagu M: Antagonistic effects of abscisic acid and jasmonates on salt stress-inducible transcripts in rice roots. Plant Cell 1997, 9: 2243–2259).

- **Thank you for the reference. This suggestion seems reasonable and will need to be considered as a hypothesis for a future root response study.**

Lines 368-369: The authors on estimating the insertion time of LRTs highlight that they used a 50% lower substitution rate as that reported in the CATD paper. Would the authors expand their choice of a lower substitution rate despite the conclusions of their results being similar to the CATD paper or why they believe that the earlier substitution rate used in the CATD paper was high hence their choice of a lower rate?

- **We made this choice independently from the CATD study, and based on Gaut et al. 1996. which shows that the substitution rate is lower in palms than in grasses. It happens that the estimated insertion ages are broadly spread around the mean value. As a result, even a ½ ratio makes little difference in practice as both articles agree on the fact that this TE burst was relatively recent and took place in a period when the coast line fluctuated to a large extent due to climatic changes (ice ages).**

Methods:

Lines 587 : Needs a re-write for clarity.

Lines 594-610: The differential gene expression screening using Noiseq method would be better presented as a flow diagram.

- **This part was completely rewritten in a “human readable” form.**

Kind regards,

Paul

Dr Paul Visendi Muhindira

REVIEWERS' COMMENTS:

Reviewer #1 (Remarks to the Author):

Summary of the key results

=====

See my previous revision.

Overall evaluation

=====

First of all I would like to thank the authors for their efforts to resolve my concerns. They have resolved most of the concerns that I raised in the first revision. Nevertheless, I still think that there are some unresolving issues.

The anchoring is poor compared with other articles in the same field. The authors wrote in their rebuttal letter "The admittedly large missing portion (53%) of the genome represents only 23% of the genes." A 23% of the gene space is almost one quarter of the genome that is a considerable amount. Based on my experience, this imposes a serious limitation on the use of this genome assembly for QTL studies because you have a 1 in 4 chance that your candidate gene is not in an anchored sequence. The authors mentioned that Hi-C and optical mapping experiments are in their way, so I am not sure if at this point it is more convenient to add these experiments to this manuscript to make it stronger. Looking into the Supplement, Fig S1 looks like LG15 should be split into two LGs.

I still think that the genome version that the authors present has more missing genes than the Latigan genome draft. This could explain the mapping percentages for the RNASeq data (~86%) but as the authors mentioned, maybe other analysis should be done.

Looking into it, I think that the new introduction facilitates the understanding of the rest of the manuscript.

About the section names, I still think that the authors should call "Results", "Results and Discussion" and "Discussion" "Conclusion". Even the guidelines of Communication Biology instruct to have two separate sections, clearly the authors discuss the results in the "Results" section and summarize the work in the "Discussion" section.

Finally, the material and methods descriptions have been improved.

Overall, I still think that the part of the genome anchoring looks incomplete to me, but it is up to the editor's decision to accept the manuscript, rewarding the considerable authors' efforts invested in this manuscript.

Reviewer #2 (Remarks to the Author):

The manuscript presents a chromosome level assembly of the coconut genome scaffolded using a linkage map using GBS. The authors further characterize the evolution of this genome highlighting close synteny with date palms.

Inference is made on the possible ancestral genome of the coconut based on ancestral genome reconstruction, and the possible role of transposable elements in the evolution of coconuts. An estimate of the time of TE insertions and whole genome duplication events are also provided. Overall,

the paper is well written, with analytical rigour. Results presented will be of great interest to coconut researchers and plant biologists in general.

The authors have satisfactorily addressed review comments in their rebuttal.

Response to reviewers

Reviewer #1 (Remarks to the Author):

Summary of the key results

=====

See my previous revision.

Overall evaluation

=====

First of all I would like to thank the authors for their efforts to resolve my concerns. They have resolved most of the concerns that I raised in the first revision. Nevertheless, I still think that there are some unresolving issues.

The anchoring is poor compared with other articles in the same field. The authors wrote in their rebuttal letter “The admittedly large missing portion (53%) of the genome represents only 23% of the genes.”. A 23% of the gene space is almost one quarter of the genome that it looks like a considerable amount. Based on my experience, this imposes a serious limitation on the use of this genome assembly for QTL studies because you have 1 possibility in 4 that your candidate gene is not in an anchored sequence.

- We are fully conscious of the limitations of the current sequence and we already acknowledged them in the article (section “Scaffold anchorage onto the map and quality assessment”). This was further made explicit in the discussion: “*The present assembly is an important milestone toward a comprehensive representation of the coconut genome. While the gene rich portion of the chromosomes (euchromatin) is faithfully represented, low recombination rates in the repeat rich portion (heterochromatin) prevented its complete and accurate assembly. Methods such as Hi-C and optical mapping will be needed to enrich it*”. This being said, the statistics we present are quite comparable to the currently available reference sequence of *Elaeis guineensis*, in terms of percentage of the assembled genome as well as in terms of proteins (but oil palm has less repeat sequences, resulting in a much less fragmented assembly!). See <https://www.ncbi.nlm.nih.gov/genome/2669> (for the sake of completeness, an improved oil palm sequence version was published this month by Ong *et al.* : *Plants* 2020, 9, 1476; doi:10.3390/plants9111476 and should become available soon.)

The authors mentioned that Hi-C and optical mapping experiments are in their way, so I am not sure if at this point it is more convenient to add these experiments to this manuscript to make it stronger.

- We would prefer not, because, even if imperfect, the current assembly represents crucial genomic information that would be extremely useful to the community of coconut geneticists and breeders. We do have a preliminary Hi-C assembly, but it needs further refining, possibly representing months of work. The current assembly will be used to supervise this revision. In fact, while linkage is liable of many small

imperfections, Hi-C may introduce a quite a few much more serious errors (i.e. chimerical assembly).

Looking into the Suppl. Fig S1 looks like the LG15 should be split into two LGs.

- What you see in this plot (now figure S2 of supplementary information) is the result of a 7 cM gap, which was not sufficient to prevent us assembling the chromosome. It is due to the presence of a gene involved in pollen competition. Furthermore, our assembly of chromosome 15 is fully supported by synteny with chromosome 9 of oil palm. The list of unassembled coconut scaffolds which most probably cover this gap is given in Table S1 of Supplementary Data 3.

I still think that the genome version that the authors present has more missing genes than Latigan's genome draft. This could explain the mapping percentages for the RNASeq data (~86%) but as the authors mentioned maybe other analysis should be done.

- We certainly agree with you on the last sentence.

Looking into I think that the new introduction facilitate the understanding of the rest of the manuscript.

About the section names, I still think that the authors should call "Results", "Results and Discussion" and "Discussion" "Conclusion". Even the guidelines of Communication Biology instruct to have two separate sections, clearly the authors discuss the results in the "Results" section and summarize the work in the "Discussion" section.

- Done

Finally the material and methods descriptions have been improved.

Overall, I still think that the part of the genome anchoring look incomplete to me, but it is up to the editor decision accept the manuscript rewarding the considerable authors' efforts invested in this manuscript.

Reviewer #2 (Remarks to the Author):

The manuscript presents a chromosome level assembly of the coconut genome scaffolded using a linkage map using GBS. The authors further characterise the evolution of this genome highlighting close synteny with date palms.

Inference is made on the possible ancestral genome of the coconut based on ancestral genome reconstruction, and the possible role of transposable elements in the evolution of coconuts. An estimate of the time of TE insertions and whole genome duplication events are also provided. Overall, the paper is well written, with analytical rigour. Results presented will be of great interest to coconut researchers and pant biologists in general.

The authors have satisfactorily addressed review comments in their rebuttal.